# UNRE: ZERO-SHOT LLM UNLEARNING VIA DYNAMIC CONTEXTUAL RETRIEVAL

## ABSTRACT

Inference-time machine unlearning with only the forget data, also known as zero-shot unlearning, is becoming increasingly important for bias mitigation, privacy preservation, copyright protection, etc. Most approaches in this domain focused on query updating, decoder modification, offline module training, or reverse-generation by the forget data. Recent works found that providing offline-prepared contexts can realize in-context unlearning. However, leveraging dynamic context (conditioned on real-time queries) to achieve zero-shot unlearning has not yet been explored, which has the potential to enforce context unlearning while preserving the performance of the original LLM. In this paper, we propose UNRE, a novel unlearning framework for LLMs that employs dynamic contextual retrieval from retrieval-augmented generation (RAG) while only leveraging the forget data. Specifically, UNRE dynamically updates contexts to guide the unlearning process in a zero-shot unlearning setting. During the inference, the user query is first leveraged for online membership inference to identify a query-specific forget set. Using this set, UNRE refines the embeddings of the retrieved chunks via gradient descent, producing adaptive contexts that steer the LLM toward a query-specific unlearned distribution. We evaluate UNRE on multiple unlearning benchmarks and show that UNRE not only outperforms existing zero-shot and context-based unlearning approaches, but also better preserves the original model performance.

## 1 INTRODUCTION

Machine unlearning is the process of revoking or forgetting data embedded in the memory of a pre-trained model (Bourtoule et al., 2021). Unlike catastrophic forgetting (Goodfellow et al., 2013), which arises unintentionally during training, machine unlearning aims to deliberately and controllably erase specific knowledge from a model. Effective unlearning is critical for building trustworthy large language models (LLMs), as it enables the removal of harmful responses (Yao et al., 2024a; Li et al., 2024; Barrett et al., 2023), copyrighted content (Dou et al., 2025; Chen et al., 2023), societal biases (Motoki et al., 2024; Yu et al., 2023), hallucinations (Yao et al., 2024a), and supports timely safety alignment (Song et al., 2025). Traditional machine unlearning methods can be categorized into targeted and untargeted approaches (Yuan et al., 2025). These methods typically require not only a *forget set*—the data to be removed from the model—but also either a reference model (Ji et al., 2024) or a *retain set*, i.e., the original training data excluding the *forget set*. The retain set can be constructed through membership inference (Shokri et al., 2017), reverse generation from the *forget set* (Pawelczyk et al., 2024), and related techniques. However, since the retain set is often unavailable in real-world scenarios (Li et al., 2024), recent works such as FLAT (Wang et al., 2025b) have been proposed to enable unlearning using only the *forget data*. Zero-shot unlearning has emerged as a scenario where the source training data is unavailable (Chundawat et al., 2023; Foster et al., 2024; Chen et al., 2025; Ahmed et al., 2025); instead, the method only requires the forget request data.

LLM unlearning targets the removal of knowledge in a designated *forget set* while preserving the model performance on other tasks (Wang et al., 2025b). Beyond data-based approaches described above, other methods include model-based unlearning, which relies on fine-tuning (Yao et al., 2024a) or training specific modules (Bhaila et al., 2025), and input-based unlearning (Liu et al., 2024a; Pawelczyk et al., 2024). Input-based methods (Liu et al., 2024a) achieve unlearning by modifying the prompt (e.g., gradient-based updates of prompt embeddings (Bhaila et al., 2025; Liu et al.,

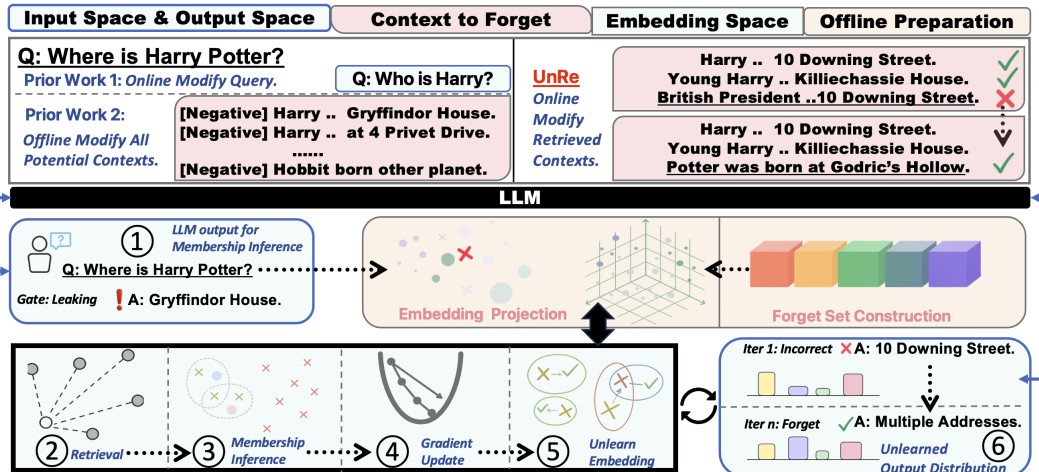

Figure 1: **Upper section: visualizing dynamic modification of the context by UNRE.** Unlike prior works which applyied a fixed *forget set* context in the prompt, or modifyied the query embeddings from original query to unlearn query, UNRE iteratively modifies query-related *forget subset* context embeddings from *forget piece* into *unlearn guiding context*(underlined). **Lower section: workflow of UNRE.** The circled step numbers show the UNRE modification progress, where step 2 to 5 happens entirely in the embedding space. Solid arrows represent communication between different components, and dashed arrows indicate state changes within the same component. The embedding model and LLM are off-the-shelf components. The legends at the top show where operations happen.

2024a)) to steer the LLM toward an unlearned output distribution (Wang et al., 2025b). Since the prompt encompasses all information provided to the model (Brown et al., 2020), inference-time unlearning, which is exemplified by input-based methods, operates during LLM inference with frozen weights and is generally regarded as suppression-intended unlearning (Ren et al., 2025). Several studies have also explored *query-adaptive* dynamic unlearning, for example, by leveraging a pretrained reference model for real-time logit-difference computation (Ji et al., 2024), by applying inference-time prompt editing through pretrained rewrite agents (Sanyal & Mandal, 2025), or by modifying the decoding process (Deng et al., 2025; Gao et al., 2025). More recent work has introduced In-Context Unlearning (ICUL) (Pawelczyk et al., 2024; Takashiro et al., 2025), which highlights context manipulation as a new perspective within input-based unlearning, enabling preservation of LLM capabilities by retaining the original query, model weights, and architecture (Takashiro et al., 2025). Meanwhile, unlearning requests will update frequently thus continual unlearning ability is important. (Gao et al., 2025)

However, existing studies have not yet explored zero-shot LLM unlearning through contextual mechanisms, and particularly the query-adaptive dynamic contexts. This gap is important: in real-world dynamic settings such as privacy protection and bias mitigation, practitioners typically only have access to the *forget set*. When unlearning is required at inference time, lightweight methods that can rapidly adapt to changes in the *forget set* are especially valuable. Moreover, such approaches offer the additional advantage of not requiring any modification to the model parameters.

In this work, we present UNRE, a RAG-based method that refines query-retrieved contexts via gradient updates in the embedding space, and leverages these adapted contexts to guide the LLM toward generating outputs aligned with the unlearned distribution. The overview of the method is shown in Figure 1. Our main contributions are summarized below:

- We introduce UNRE, to the best of our knowledge, the first framework to leverage query-specific dynamic contexts for achieving zero-shot unlearning.

- We develop an online membership-inference–guided RAG architecture that first identifies the query-specific unlearning target, then optimizes the retrieved contexts for unlearning generation, thereby minimizing disruption to the LLM's original capabilities.

- UNRE is 100% pre-processing free and query-adaptive, which better aligns with the dynamic requirements of real-world scenarios.

- Through extensive experiments on diverse unlearning tasks across multiple LLMs, UNRE demonstrates superior unlearning effectiveness, e.g., around 3 times stronger than fixed contextual unlearning, while largely preserving the model's original performance by maintaining a similar perplexity score as the original model.

## 2 RELATED WORKS

**Machine Unlearning**   Machine unlearning aims to remove specific behaviors or knowledge without broadly degrading model utility (Cao & Yang, 2015). Liu et al. (2024b) formulates the unlearning target as a confounder between an LLM's input and output, and casts unlearning as a deconfounding process. Recent works have explored loss adjustment using only the *forget set*, thereby avoiding reliance on retain data or auxiliary agents (Wang et al., 2025b; Yao et al., 2024a). Zero-shot unlearning was introduced as a scenario where only the forget data is available (Foster et al., 2024). For instance, Gu et al. (2025) proposed to generate an anti-forget set to enhance fine-tuning–based unlearning. Other approaches, including PROD (Jiang et al., 2025), DPO (Rafailov et al., 2023), and NPO (Zhang et al., 2024), constrain unlearning with original model outputs to preserve overall performance. On the other hand, CEU (Entesari et al., 2025) flattens the forget set logits while enforcing a retain set performance lower bound to preserve model utility after tuning. In this work, we adopt a related loss-based formulation but operate solely on the *forget set*, eliminating the need for the retain data or auxiliary models.

**Inference-time Unlearning**   Prompt engineering has emerged as a method for performing unlearning at inference time. For example, SPUL (Bhaila et al., 2025) trains soft prompts during an offline stage using a pre-established forget set and retain set, guiding LLMs to generate outputs that approximate a machine-unlearned distribution. *ECO* (Liu et al., 2024a) trained a classifier for unlearn-required prompt offline and a general corruption parameter that is applied to vectorized user input query in the embedding space to guide LLM to generate output in an unlearned distribution. Contrastive decoding methods, such as UCD and ULD (Suriyakumar et al., 2025; Ji et al., 2024), leverage logit differences between a small model trained on Forget and Retain Sets to guide unlearning. Since providing context adaptive to a specific query will make the LLM perform In-Context Learning (ICL) differently (Garg et al., 2022), and context can be updated by gradient in the embedding space during inference time (Zou et al., 2023), In-Context Unlearning (ICUL) (Pawelczyk et al., 2024) uses prompt context constructed from Forget and Retain Sets to prevent the generation of unwanted content. Vector steering techniques (Li et al., 2023; Rimsky et al., 2024; Arditi et al., 2024; Cao et al., 2024; Dunefsky & Cohan), including *InferAligner* (Wang et al., 2024a) and *FairSteer* (Li et al., 2025), inject offline-prepared steering vectors into LLM layers to influence inference. Other methods modify the decoder or employ multi-agent systems for unlearning (Wang et al., 2025a; Deng et al., 2025; Sanyal & Mandal, 2025). All these approaches, however, require either a retain set or offline training/tuning.

**Retrieval Augmented Generation and Unlearning via RAG**   Retrieval-Augmented Generation (RAG) has seen significant advances in recent years, improving LLM performance by providing relevant external information during generation (Lewis et al., 2020b). A typical RAG pipeline involves chunking, embedding, retrieval, and reranking (Lewis et al., 2020a), and recent methods have focused on better aligning the retriever with the LLM. For example, *REPLUG* (Shi et al., 2024) tunes the retriever based on LLM's likelihood signal to update the retrieval embeddings via gradient, which improves both perplexity and downstream accuracy. In this work, we adopt a multi-query RAG framework (Cheng et al., 2024) in which the embeddings are aligned with the LLM and receive gradient updates from its outputs, enabling more effective and adaptive retrieval during generation.

Several recent works have explored using RAG for unlearning at inference time. Wang et al. (2024b) constructs a retain set from the forget set offline and injects it into RAG for unlearning. *Eraser4RAG* (Wang et al., 2025d) trains a rewrite agent via reinforcement learning to transform retrieved forget data into retainable content, while *De-Indexing* (Vilella & Ruffo, 2025) reranks retrieved items to promote the retain set over the forget set. Similar to other prior works on inference-time unlearning, all these approaches require either a preprocessed retain set or prior agent training, limiting their applicability in scenarios where only the forget set is available.

# 3 METHODOLOGY

In this work, we propose an inference-time zero-shot unlearning framework UNRE based on RAG that operates solely with the *forget set*, requiring neither additional training nor fine-tuning throughout the workflow nor any architectural modifications to the LLM.

## 3.1 PROBLEM STATEMENT

The goal of query-adaptive zero-shot unlearning is to force the targeted LLM $\mathbf{M}$ to generate an output $y_u$ in an unlearned token-sequence distribution, given only the *forget set* $\mathbf{O}$, where $\mathbf{O} = \{(\mathbf{o}_i)\}_{i=1}^{n}$ ($i \in [1, n]$) is the $n$ chunk pieces among the *forget set* and $\mathbf{o}_i$ is a piece in the *forget set*. Each forget piece $\mathbf{o}_i = \{\mathbf{x}_i, \mathbf{y}_i\}$ contains the *feature/example input* $\mathbf{x}$ and *label* $\mathbf{y}$.

We consider a scenario where the UNRE owner has access to neither the *model training data* nor the *retain data* (in contrast to prior unlearning methods (Yao et al., 2024a)), and where the user query itself remains unaltered (Liu et al., 2024a)—with only the retrieved context being modified at inference time. To this end, the unlearning objective is to find a perturbed set $\tilde{\mathbf{O}}_q$, where $\mathbf{O}_q$ is the query-related subset of $\mathbf{O}$, so that using $\tilde{\mathbf{O}}_q$ as context to constrain the LLM inference generation progress $y_u = \mathbf{M}(q; \tilde{\mathbf{O}}_q)$.

## 3.2 METHOD OVERVIEW

We propose UNRE framework to find the proper $\tilde{\mathbf{O}}_q$. The overview of the method is shown in Figure 1. During the offline stage, $\mathbf{O}$ will be input into the RAG, a technique that augments LLM generation through retrieving query-related contexts, stored in the RAG embedding vector database $\mathbf{V}_R$ (steps 1 to 2 in Figure 1), which can be aligned with LLM embeddings $\mathbf{V_M}$ (Cheng et al., 2024).

During inference, UNRE consists of the following stages. First, online membership inference (He et al., 2025; Fu et al., 2024) for $\mathbf{O}_q$ (steps 1 to 2 in Figure 1). When receiving the user query $q$, the query will go through the LLM, generating a regular output $y_q$. Then, the input query $q$ and output $y_q$ will be sent to a RAG retrieval module, which will conduct a similarity search in $\mathbf{V}_R$ (step 3 in Figure 1). If the similarity result is higher than a *threshold gate* $\tau$, unlearning is required. Second, a dynamic unlearned context updating process for $\tilde{\mathbf{O}}_q$. This is achieved through gradient descent inside embedding space, aiming at optimizing the *unlearning loss function* (steps 4 to 5 in Figure 1) by using a perturbation matrix $\delta$, which constrains the LLM output $y_u = \mathbf{M}(q; \tilde{\mathbf{O}}_q)$ (step 6 in Figure 1) into an unlearned distribution. Note that $\phi$ is the original embedding model (such as e5).

## 3.3 PRE-CHECK GATE WITH ONLINE MEMBERSHIP INFERENCE

The pre-check progress aims to minimize the UNRE influence on the model's original performance by shutting down the UNREwhen not needed. We first obtain a regular output $y_q = \mathbf{M}(q)$. By following a standard retrieval process in RAG (Lewis et al., 2020a), we compute its retrieval similarity to the *forget set* $\mathbf{O}$, as follows:

$$\max_{i \in [n]} \text{Similarity}\big(\phi(y_q),\, \phi(o_i)\big) \;<\; \tau. \tag{1}$$

The similarity threshold $\tau$ is a user-defined parameter, and $\phi$ is the embedding progress. We use *L2 distance in embedding* for similarity calculation. If the similarity is below $\tau$, $y_q$ is returned; otherwise, the UNRE pipeline starts.

## 3.4 UNLEARN PREPARATION OF UNRE

Thus, we have a real-time, query-specific *forget subset* $\mathbf{O}_q$ through the retrieval progress,

$$\mathbf{O}_q \;=\; \big\{\, o_i \in \mathbf{O} \;:\; -d\big(\phi(y_q),\, \phi(o_i)\big) \;\geq\; \tau \,\big\}. \tag{2}$$

where $d$ is the L2 distance between embedding vectors; and thus we have the embedding vector of sub-*forget set* $E_O = \phi(\mathbf{O}_q)$. Through the retrieval similarity search during $\phi(y_q)$, we obtain the

---

**Algorithm 1** Gradient-based Update in UNRE Embedding

---

**Require:** query $q$; LLM $\mathbf{M}$; matrixed forget context $v_1$; pre-set budget $\varepsilon$; pre-set step size $\eta$; gradient update steps $j \in [1, J]$; query-specific unlearn matrix $\delta$; token position $t \in [1, T]$ in a sentence; last-layer hidden state $h_\delta(t)$; last-layer logits $z_\delta(t)$

1: $\delta^{(0)} \leftarrow \mathbf{0}$; $y_q \leftarrow \mathbf{M}(q)$
2: *Get* $h_0(t)$, $z_0(t)$ while $\mathbf{M}$ generating $y_q$
3: **for** $j = 1$ **to** $J$ **do**
4:      $\tilde{v}_1 \leftarrow v_1 + \delta^j$;
5:      LLM output: $y_u \leftarrow \mathbf{M}(q \oplus \tilde{v}_1)$
6:      *Get* $h_\delta(t)$, $z_\delta(t)$ while $\mathbf{M}$ generating
7:      *Calculate* $\bar{h}_\delta \leftarrow \frac{1}{T} \sum_{t=1}^{T} h_\delta(t)$,
8:      Semantic Similarity: $S \leftarrow \text{Similarity}\big(\bar{h}_\delta, \phi(y_q)\big)$
9:      *Calculate* $\hat{z}_\delta(t)$ given $z_\delta(t)$; $\hat{z}_0(t)$ given $z_0(t)$
10:     Next-Token Distribution Shift: $\mathcal{N} \leftarrow \frac{1}{T} \sum_{t=1}^{T} \cos\big(\hat{z}_\delta(t), \hat{z}_0(t)\big)$
11:     $\mathcal{L} \leftarrow \text{softplus}\big(\mathcal{N} - S\big)$
12:     *Gradient Update* $\delta^{j+1} \leftarrow \text{PGD}(\mathcal{L})$
13: **end for**
14: **return** $\delta^J$

---

initial RAG-retrieved, query-related embedding $E_R$,

$$E_R = \text{TopK} \big( - d(\phi(y_q), \phi(o_i))\big), \tag{3}$$

where $K$ is the RAG retrieval parameter defined by the user (Lewis et al., 2020b).

We obtain $E_1 = E_O \cup E_R$, and then form $E_1$ into matrix $v_1$. Prior works like *PromptReps* and *HyDE* (Gao et al., 2023; Zhuang et al., 2024) have shown that looping back LLM output can enhance the retrieval progress. Starting from the retrieved query-related context and limiting the context example amount $K$ can retain more of LLM's original performance (Pawelczyk et al., 2024).

### 3.5 UPDATE EMBEDDING VECTORS TO UNLEARN IN UNRE

#### 3.5.1 DESIGN OF LOSS FUNCTION FOR THE UNLEARNING OBJECTIVE

We optimize a perturbation matrix $\delta$ and feed the perturbed input $\tilde{v}_1 = v_1 + \delta$ into $\mathbf{M}$. The method is detailed in Algorithm 1. In our setting, we perturb (applying $\delta$ to) the groundtruth $y_i$ — the specific content that must be forgotten — following ICUL (Pawelczyk et al., 2024), which applies label reversal during its offline stage to enhance forgetting.

At each output token position $t \in \{1, \ldots, T\}$, the model outputs a last-layer hidden state $h_\delta(t)$ and logits $z_\delta(t)$ Yao et al. (2024a). $\oplus$ in line 5 represents the progress of projecting the embedding vector back to discrete tokens and combining it with the current text. We start at $\delta = \mathbf{0}$, where we have $h_0(t)$, $z_0(t)$, as shown in lines 1 and 2 in Algorithm 1. The design of the UNRE loss is motivated by maintaining the *semantic meanings* while increasing *token distributional shift* (Sinha et al., 2025; Liu et al., 2024b; Wang et al., 2025c) of unlearned output $y_u$, as discussed below.

**Sentence semantics** For Semantic Similarity $S$, as illustrated in lines 7 to 8 of the Algorithm 1, we aggregate hidden states into a sentence vector and compute semantic similarity (higher is better).

**Distributional shift of next-token predictions.** As presented in lines 9 to 10 of Algorithm 1, we let $\hat{z}(t) = z(t)/\|z(t)\|_2$ denote the unit-direction of logits. We define $\mathcal{N}$, the expectation of the token-level directional discrepancy (lower is better). Consequently, we have the *loss function* (line 11 of Algorithm 1):

$$\mathcal{L}(\delta) = \text{softplus}\big(\mathcal{N} - S\big) = \log\big(1 + \exp(\mathcal{N} - S)\big) \tag{4}$$

Since loss adjustment can flexibly realize diverse unlearning objectives (Wang et al., 2025b), we generalize the loss function $\mathcal{L}$ to support a broad range of unlearning tasks (e.g., copyright, privacy) by tuning task-specific parameters and integrating the ECO loss formulation (Liu et al., 2024a), as detailed in Appendix E.1.2.

### 3.5.2 GRADIENT-BASED UPDATE IN CONTEXT EMBEDDING

We employ Projected Gradient Descent (PGD) (Madry et al., 2018) to update gradients in the embedding space, while constraining the update region to avoid the *forget set* embedding $e = \phi(O)_q$. Specifically, we optimize the perturbation $\delta$ (line 10 in Algorithm 1) to minimize the loss $\mathcal{L}$.

$$\delta^{(j+1)} = \Pi_{\{\delta:\ \|\delta\|_2 \leq \varepsilon,\ \min_{e \in E} \|v_1 + \delta - e\|_2 \geq \tau\}} \left( \delta^{(j)} - \eta \nabla_\delta L(\delta^{(j)}) \right), \tag{5}$$

where $\delta^{(j)}$ has the same dimension as $v_1$, denotes the query-specific unlearning perturbation matrix at PGD step $j$; $\nabla_\delta \mathcal{L}(\delta^{(j)})$ is the gradient of the loss $\mathcal{L}$ evaluated at $\delta^{(j)}$; $\eta$ is the learning rate for the gradient update; $\Pi$ is the projection operator onto the perturbation ball with budget $\varepsilon$ specified by the UNRE owner; $j$ is the iteration index; and $J$ is the total number of PGD steps.

### 3.6 UNRE UNLEARNING INFERENCE

After obtaining $\delta^J$ from Algorithm 1, we construct the perturbed matrix $v_c = v_1 + \delta^J$ for final inference. UNRE then constrains the LLM using updated contexts $C = \tilde{\mathbf{O}}_q$ decoded from $v_c$, thereby guiding the model to generate unlearned outputs $y_u = \mathbf{M}(q; C)$.

Finally, the pre-check procedure described in Section 3.3 is started again to determine whether another run of UNRE is necessary, ensuring that the final LLM output $y_u$ exhibits no similarity to the *forget set* $\mathbf{O}$.

## 4 EXPERIMENT

**Overview** In this section, we evaluate UNRE across a range of unlearning tasks, including *Entity Unlearning* and *Copyright Content Unlearning*, using the *TOFU* (Maini et al., 2024), *RWKU* (Jin et al., 2024), and *HP* (Eldan & Russinovich, 2023) datasets. We further assess its performance on *context unlearning* under varying context lengths, comparing against a state-of-the-art in-context unlearning method (Pawelczyk et al., 2024). Additional tasks and results are provided in Appendix. All experiments are conducted on Nvidia L40S GPUs.

**Baseline Methods** We compare UNRE against a diverse set of unlearning baselines, grouped into three categories. *Gradient-based methods* include Gradient Ascent (**GA**) (Maini et al., 2024), Grad-Diff (**GD**)(Maini et al., 2024), KL minimization (**KL**)(Maini et al., 2024), Large Language Model Unlearning (**LLMU**) (Yao et al., 2024a), and **Mismatch** (Yao et al., 2024b), as well as regularized GA variants **GAGDR** and **GAKLR** (Shi et al., 2025). *Preference-based methods* include Preference Optimization (**PO**) (Maini et al., 2024), Direct Preference Optimization (**DPO**) (Maini et al., 2024), Negative Preference Optimization (**NPO**) (Zhang et al., 2024), and the regularized NPO variants **NPOGDR** and **NPOKLR** (Shi et al., 2025), together with the forget-only loss-adjustment method **FLAT** (Wang et al., 2025b). *Tuning-free methods* include In-Context Unlearning (**ICUL**) (Pawelczyk et al., 2024), **ECO** (Liu et al., 2024a), **GUARD** (Deng et al., 2025), and **Prompt**/Output-Filtering strategies (Deng et al., 2025; Pawelczyk et al., 2024). We include more baselines and their descriptions in Appendix.

### 4.1 ENTITY UNLEARNING

#### 4.1.1 TOFU 1% SPLIT

We evaluate entity unlearning on the **TOFU 1% Split** benchmark (Maini et al., 2024) Following prior work, we first fine-tune each base LLMs on the full TOFU training set to obtain the *Original LLM*; the *Retained LLM* is fine-tuned on the split, which serves as the reference model. We report the 1% forget split and use LLMs of Falcon3-7B, Llama3.2-3B and Qwen2.5-7B, as summarized in Table 1.

**Metrics** We adopt the official TOFU evaluation metrics. **Forget Quality (FQ)** is defined as the $p$-value from a Kolmogorov–Smirnov test applied to the Truth Ratio distributions of the unlearned and retained models on the forget set; higher values indicate stronger unlearning performance. **Model Utility (MU)** is computed as the harmonic mean of Answer Probability, Truth Ratio, and ROUGE-L

Table 1: **TOFU 1% split**. Performance of our method and baseline methods on the TOFU dataset using three base LLMs (Falcon3-7B, Llama3.2-3B and Qwen2.5-7B). FQ, MU, F-RL, and R-RL denote *forget quality*, *model utility*, *ROUGE-L on the forget set*, and *ROUGE-L on the retain set*, respectively. We include the Original LLM and the Retained LLM (trained on retain set) for reference.

| | Falcon3-7B-Instruct | | | | Llama3.2-3B-Instruct | | | | Qwen2.5-7B-Instruct | | | |
|---|---|---|---|---|---|---|---|---|---|---|---|---|
| **Method** | **FQ**↑ | **MU**↑ | **F-RL**↓ | **R-RL**↑ | **FQ**↑ | **MU**↑ | **F-RL**↓ | **R-RL**↑ | **FQ**↑ | **MU**↑ | **F-RL**↓ | **R-RL**↑ |
| Original LLM | 0.0067 | 0.6644 | 0.8612 | 0.8030 | 0.0067 | 0.5752 | 0.9913 | 0.9778 | 0.0067 | 0.6054 | 0.9719 | 0.9219 |
| Retained LLM | 1.0 | 0.6647 | 0.3792 | 0.7998 | 1.0 | 0.6018 | 0.4088 | 0.9866 | 1.0 | 0.5910 | 0.3794 | 0.8958 |
| GA | 0.0067 | 0.6663 | 0.7379 | 0.8041 | 0.0067 | 0.5754 | 0.8112 | 0.9735 | 0.0541 | 0.5887 | 0.4723 | 0.8837 |
| KL | 0.0067 | 0.6653 | 0.7347 | 0.7943 | 0.0066 | 0.5759 | 0.8331 | 0.9755 | 0.0970 | 0.5876 | 0.4613 | 0.8820 |
| GD | 0.0286 | 0.6535 | 0.7058 | 0.8195 | 0.0066 | 0.5747 | 0.8359 | 0.9771 | 0.0286 | 0.5929 | 0.4745 | 0.8848 |
| LLMU | 0.0287 | 0.6544 | 0.7589 | 0.8183 | 0.0143 | 0.5680 | 0.9913 | 0.9765 | 0.0286 | 0.5656 | 0.4774 | 0.5823 |
| PO | 0.0067 | 0.6625 | 0.8290 | 0.8084 | 0.0143 | 0.5678 | 0.9913 | 0.9774 | 0.0067 | 0.6152 | 0.7387 | 0.8459 |
| DPO | 0.0286 | 0.6535 | 0.7058 | 0.8195 | 0.0065 | 0.5766 | 0.7379 | 0.9769 | 0.0067 | 0.5766 | 0.7379 | 0.5259 |
| NPO | 0.0067 | 0.6656 | 0.7432 | 0.7958 | 0.0067 | 0.5768 | 0.7866 | 0.9765 | 0.0143 | 0.5539 | 0.4055 | 0.5258 |
| FLAT | 0.0030 | 0.6659 | 0.7013 | 0.7994 | 0.0066 | 0.5766 | 0.7379 | 0.9769 | 0.0286 | 0.5971 | 0.5079 | 0.9032 |
| ICUL | 0.0286 | 0.6641 | 0.4059 | 0.8028 | 0.0143 | 0.5751 | 0.5614 | 0.9778 | 0.0143 | 0.6054 | 0.4539 | 0.9217 |
| Prompt | **0.0970** | 0.6644 | 0.4045 | 0.8030 | 0.0143 | 0.5753 | 0.8635 | 0.9777 | 0.0067 | 0.6053 | 0.5552 | 0.9218 |
| GUARD | 0.0541 | 0.6643 | 0.3115 | 0.8029 | 0.5786 | 0.5752 | 0.3764 | 0.9776 | 0.2656 | 0.6052 | 0.3691 | 0.9219 |
| **UnRe (Ours)** | 0.0611 | 0.6644 | **0.2824** | 0.8030 | **0.6012** | 0.5752 | **0.3298** | 0.9778 | **0.2977** | 0.6054 | **0.3169** | 0.9219 |

across the subsets retain, real authors, world facts, where higher scores reflect better utility preservation. We also report **F-RL** (ROUGE-L on the forget set; lower is better) and **R-RL** (ROUGE-L on the retain set; higher is better).

**Results** It can be seen that UNRE demonstrates strong unlearning performance while preserving model utility across modern LLMs. On Llama3.2-3B and Qwen2.5-7B, it achieves model utility (MU) scores of 0.5752 and 0.6054, staying within 0.28% and 1.6% of the best train-time baselines (Original/ICUL/GUARD). At the same time, UNRE attains superior Forget Quality (FQ), reaching 0.6012 on Llama3.2-3B and 0.2977 on Qwen2.5-7B, surpassing GUARD, while remaining competitive on Falcon3-7B (0.0611) and outperforming gradient-based baselines. Across all models, it maintains a favorable forget–retain trade-off, achieving the lowest forget–retain loss (F-RL) while keeping retain–retain loss (R-RL) at the Original level. Overall, UNRE provides effective unlearning with minimal impact on model utility across different LLMs.

### 4.1.2 REAL-WORLD KNOWLEDGE UNLEARNING (RWKU)

We also evaluate entity unlearning on the **RWKU** benchmark (Jin et al., 2024), as a *test-only* suite. The *Original LLM* (Before) denotes the base model without unlearning, and our method is applied at inference time using a forget-only retrieval corpus derived from RWKU materials. We report results on LLaMA-3-8B-Instruct and LLaMA-3.1-8B-Instruct, as presented in Table 2.

**Metrics** We adopt the official RWKU metrics. **Forget** reports ROUGE-L on Fill-in-the-Blank and QA probes over the forget targets (FB/QA; lower is better); **AA** denotes adversarial probes in robustness analyses. **Neighbor** is ROUGE-L on probes about entities adjacent to the forget targets and reflects locality (higher is better). **MIA** reports membership inference on forget- and retain-like samples via **FM** (higher is better) and **RM** (lower is better). **Utility** measures general capabilities on reasoning, truthfulness, factual QA, and fluency (**Rea**, **Tru**, **Fac**, **Flu**; higher is better).

**Results** Across both LLMs, UNRE delivers the strongest *forgetting* while preserving *locality*, *privacy*, and *utility*. On LLaMA-3-8B, it reduces **Forget–QA** to 39.8, outperforming most baselines, and increases **Neighbor–QA** to 78.1 (vs. 76.5 for GAGDR, the base baseline), indicating reduced collateral forgetting. For **MIA**, UNRE achieves higher **FM** (268.7, above NPO as the best baseline) and lower **RM**, reflecting weaker membership signals on the forget set and fewer false positives on *retain-like data*. **Utility** is maintained showing minimal degradation to general model capabilities. On LLaMA-3.1-8B, UNRE further lowers **Forget–AA** to 38.7, and improves **Neighbor–FB** to 74.0 (surpassing best baseline NPO$_{GDR}$). For **MIA**, it achieves higher **FM** and the best **RM**, reflecting effective unlearning without falsely flagging *retain data*. Model **Utility** remains robust for LLaMA-3.1-8B as well, with overall trends comparable to baseline performance.

Table 2: **RWKU.** We report *Forget* (FB/QA/AA/All, ↓), *Neighbor* (FB/QA/All, ↑), *MIA* (FM↑/RM↓), and *Utility* (Rea/Tru/Fac/Flu, ↑).

(a) LLaMA-3-8B-Instruct

| | Forget↓ | | | Neighbor↑ | | MIA | | Utility↑ | | | |
|---|---|---|---|---|---|---|---|---|---|---|---|
| Method | FB | QA | AA | FB | QA | FM↑ | RM↓ | Rea | Tru | Fac | Flu |
| Before | 85.6 | 70.3 | 74.7 | **93.1** | **82.0** | 236.5 | 230.9 | 41.0 | 36.4 | 53.7 | 704.6 |
| GA | 72.0 | 64.6 | 68.5 | 85.0 | 74.7 | 241.4 | 234.6 | 40.4 | **37.6** | 49.6 | **710.3** |
| GAGDR | 72.6 | 64.0 | 69.7 | 86.2 | 76.5 | 242.8 | 236.8 | 39.6 | 36.8 | 50.4 | **710.3** |
| GAKLR | 70.7 | 57.5 | 69.9 | 80.5 | 70.5 | 242.4 | 230.8 | **41.5** | 35.6 | 54.0 | 704.4 |
| NPO | 46.6 | **39.0** | 35.3 | 79.2 | 70.9 | 263.3 | 241.4 | 40.5 | 36.0 | **56.7** | 695.9 |
| NPOGDR | 52.2 | 43.9 | 42.9 | 82.5 | 70.5 | 254.5 | 240.1 | 39.6 | 37.2 | 51.4 | 708.2 |
| NPOKLR | 52.5 | 40.6 | 43.2 | 83.2 | 72.1 | 253.0 | 236.9 | 40.9 | 35.4 | 54.2 | 704.9 |
| **UnRe (Ours)** | **44.8** | 39.8 | **34.9** | 88.4 | 78.1 | **267.7** | 236.2 | 40.6 | 36.0 | 53.7 | 704.6 |

(b) LLaMA-3.1-8B-Instruct

| | Forget↓ | | | Neighbor↑ | | MIA | | Utility↑ | | | |
|---|---|---|---|---|---|---|---|---|---|---|---|
| Method | FB | QA | AA | FB | QA | FM↑ | RM↓ | Rea | Tru | Fac | Flu |
| Before | 63.9 | 65.1 | 69.5 | **74.1** | 69.8 | 223.5 | **218.2** | 42.2 | 35.4 | 61.2 | 695.2 |
| GA | 50.7 | 45.4 | 61.2 | 45.6 | 37.2 | **248.9** | 241.9 | 43.2 | **35.8** | 48.7 | **726.6** |
| GAGDR | 55.4 | 49.6 | 63.9 | 60.2 | 53.5 | 239.8 | 231.3 | **44.2** | 35.0 | 53.9 | 718.5 |
| GAKLR | 62.7 | 49.9 | 66.4 | 67.9 | 61.2 | 235.8 | 223.0 | 42.6 | 35.4 | 59.0 | 682.1 |
| NPO | **35.7** | 40.2 | 39.0 | 67.3 | 66.2 | 241.4 | 220.5 | 42.5 | 35.6 | **61.8** | 684.2 |
| NPOGDR | 42.4 | **37.2** | 42.0 | 74.0 | 66.7 | 236.3 | 220.1 | 43.0 | 35.4 | 60.8 | 698.8 |
| NPOKLR | 40.6 | 41.4 | 42.2 | 73.3 | **69.9** | 234.4 | 218.8 | 42.3 | 35.4 | 61.5 | 695.1 |
| **UnRe (Ours)** | 39.2 | 37.9 | **38.7** | 74.0 | 68.6 | 242.4 | 220.1 | 42.3 | 35.4 | 61.2 | 695.0 |

## 4.2 COPYRIGHTED CONTENT UNLEARNING

We use **Harry Potter and the Sorcerer's Stone** (Eldan & Russinovich, 2023) (HP) as copyrighted content to be forgotten, constructing **forget** and **retain** splits by extracting 400 chunks from the book for the *forget set* and sampling 400 paragraphs from C4 for the *retain set*. The LLM is fine-tuned on the *forget set* to simulate memorization, while the original pretrained checkpoint serves as the retained baseline.

**Metrics** We report the **Forget Quality Gap (FQ Gap)** defined over BLEU and ROUGE-L differences between the unlearned and the retained model on *the split forget set*, together with **Perplexity (PPL)** (Jelinek et al., 1977) and the average zero-shot accuracy (**Avg. Acc.**) across nine standard tasks as a model-utility proxy. We evaluate on OPT-2.7B and Llama2-7B models for better comparison with prior works.

**Results** It can be seen from the results that UNRE achieves effective unlearning without compromising model utility in general. In the HP setting, it consistently enforces strong forgetting while preserving general capabilities. Operating entirely at inference time, the framework activates conservatively only on copyright-relevant queries, ensuring that LLM generation text quality (PPL) and zero-shot accuracy remain aligned with the original checkpoint across architectures. This demonstrates the core goal of inference-time unlearning: *eliminate targeted knowledge while maintaining unrelated model capabilities*.

Besides, it can be observed that prior methods, which do not explicitly balance forgetting and utility, typically fail in one of two ways: (i) improving the forget score but degrading fluency or accuracy, or (ii) preserving general performance while leaving residual memorization. By contrast, UNRE successfully preserves the retained model's utility profile while removing reproduction of the copyrighted text. Prompt- or filter-based baselines largely leave non-trigger inputs unchanged and fail to provide targeted suppression, whereas optimization-based methods can achieve forgetting but often at the expense of the generation text quality. The results highlight that, as a lightweight, training-free method, UNRE effectively performs copyright-content unlearning in HP, achieving targeted knowledge removal while retaining overall generation performance, outperforming prior methods.

Table 3: **HP unlearning** on OPT-2.7B and Llama2-7B. Lower FQ Gap/PPL and higher Avg. Acc. are better.

| Method | OPT-2.7B | | | Llama2-7B | | |
|---|---|---|---|---|---|---|
| | FQ Gap ↓ | PPL ↓ | Avg. Acc. ↑ | FQ Gap ↓ | PPL ↓ | Avg. Acc. ↑ |
| Original LLM | 1.5346 | 15.6314 | 0.4762 | 3.6594 | 8.9524 | 0.5617 |
| Retained LLM | 0.0000 | 14.3190 | 0.4686 | 0.0000 | 8.7070 | 0.5599 |
| KL | 2.7301 | 16.1592 | 0.4688 | 0.4225 | 9.4336 | 0.5509 |
| GD | 2.3439 | 16.1972 | 0.4690 | 0.5304 | 9.1797 | 0.4902 |
| Mismatch | 1.4042 | 15.7507 | 0.4679 | 0.4647 | 8.9906 | 0.5593 |
| LLMU | 2.4639 | 15.8398 | 0.4656 | 0.1985 | 9.0530 | 0.5503 |
| PO | 2.1601 | 14.8960 | 0.4583 | 0.5124 | 8.8364 | 0.5532 |
| DPO | 2.2152 | 16.8396 | 0.4621 | 0.2924 | 8.9597 | 0.5614 |
| NPO | 1.2611 | 19.6637 | 0.4644 | 0.5151 | 9.0397 | 0.5609 |
| FLAT | 1.4089 | 15.5543 | 0.4686 | 0.2265 | 8.9906 | 0.5580 |
| ICUL | 1.0121 | 15.6314 | 0.4762 | 2.5585 | 8.9524 | 0.5617 |
| GUARD | 0.6314 | 15.6314 | 0.4762 | 0.1367 | 8.9524 | 0.5617 |
| **UnRe (Ours)** | **0.6112** $\pm$ **0.0011** | 15.6314 | 0.4762 | **0.1207** $\pm$ **0.0008** | 8.9524 | 0.5617 |

| Method | Accuracy ↑ | TPR @ FPR=0.01 ↓ |
|---|---|---|
| Baseline | 90.4% | 0.0267 |
| ICUL | 89.8% | 0.0179 |
| **UnRe** | 90.0 $\pm$ 0.2% | **0.0147** |

*(a) Unlearning results at $K = 10$.*

| Method | Accuracy ↑ | TPR @ FPR=0.01 ↓ |
|---|---|---|
| Baseline | 90.4% | 0.0267 |
| ICUL | 90.2% | 0.0183 |
| **UnRe** | **90.3** $\pm$ **0.2%** | **0.0153** |

*(b) Unlearning results at $K = 5$*

*(c) Unlearning performance (TPR).*

*(d) Accuracy.*

Figure 2: Evaluate unlearning for different numbers of deletion requests (1, 5, 10, 20).

### 4.3 CONTEXTUAL UNLEARNING COMPARISON FOR DIFFERENT CONTEXT LENGTHS

We follow the ICUL (Pawelczyk et al., 2024) setup and adopt its *LiRA-Forget* protocol (Carlini et al., 2022) to quantify unlearning. We evaluate inference-time unlearning across varying context lengths, considering 5 and 10 deletions (i.e., $K = 5$ and $K = 10$ retrieved context examples). As shown in Figure 2.

**Metrics**  The **TPR @ FPR=**$0.01$ measures the true positive rate of a likelihood-ratio test distinguishing an unlearned model from a retained trained model on the forget points (*lower is better*). **Accuracy** reflects standard test performance on held-out data, serving as a utility indicator (*higher is better*). Effective unlearning is indicated by TPR values approaching the benchmark while maintaining accuracy close to the baseline. **Baseline** refers to the original fine-tuned model without any unlearning.

**Results**  UNRE demonstrates the intended behavior of inference-time unlearning using in-context examples: it largely preserves task accuracy, consistently outperforming ICUL across varying context lengths and approaching the performance of *GA*. While ICUL's forgetting improves with longer contexts, its overall unlearning effectiveness remains substantially below that of UNRE.

## 5 CONCLUSION

In this work, we propose UNRE, a novel retrieval-based framework for dynamic, query-adaptive zero-shot unlearning in LLMs. Unlike prior approaches that rely on fixed prompts or static context injection, UNRE leverages query-adaptive dynamic contexts to achieve inference-time unlearning without any offline preparation. The framework first employs online membership inference to guide retrieval from the *forget set*, adapting context to each query, and then applies gradient-based perturbations to the retrieved embeddings to steer the LLM's outputs toward an unlearned distribution. Empirical results across multiple LLMs and unlearning tasks demonstrate that UNRE effectively removes targeted knowledge while preserving the model's original capabilities. Notably, it operates without pretraining or retain sets, making it particularly suitable for lightweight, real-world unlearning scenarios where the *forget set* is frequently updated. Overall, UNRE illustrates that dynamic context can enable efficient, query-adaptive zero-shot unlearning during LLM inference.

## ETHICS STATEMENT

We adhere to the ICLR Code of Ethics. UNRE is an inference-time, training-free unlearning controller that operates *only* with the forget set and leaves the base model's parameters unchanged; a conservative pre-check gate prevents activation on benign inputs. As a result, the method targets removal/suppression of copyrighted passages and hazardous knowledge while preserving general utility, thereby *reducing* potential harm rather than introducing new risks. Our experiments rely on standard public benchmarks (e.g., Harry Potter excerpts for copyright unlearning; WMDP for hazardous-knowledge attenuation) and do not involve human subjects or the collection of personal data; no copyrighted material is redistributed. We release code and prompts with safeguards aimed at preventing misuse (e.g., documentation on intended use and limitations). Overall, UNRE is designed to strengthen ethical deployment by enabling targeted forgetting without degrading unrelated capabilities.

## REPRODUCIBILITY STATEMENT

All experimental settings (datasets, splits, preprocessing, model variants, hyperparameters, training schedules, and evaluation protocols) are described in detail in Section 4. We conduct all experiments on a single node equipped with $4\times$ NVIDIA L40S GPUs. We submit the code in the supplementary material, which includes a fully specified runtime environment and scripts to reproduce results.

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

APPENDIX

## A SUMMARY OF APPENDIX

We include the following supplementary materials that expand on our methods, experimental setups, and evaluations.

- B **LLM Usage Disclosure** - We detailed how we used LLM during the conduct of this project.
- C **Hyperparameter** - We show hyperparameters we used in the experiments.
- D **Extension Literature Reviews and Preliminaries** - We provided more details on the loss function and preliminaries.
- E **Additional Details of Methodology** - We provide the details of method scalability under different tasks, alignment method, and comparison with prior RAG-based unlearning, as well as illustrating the inference workflow for better understanding.
- F **Additional Experiment Settings** - We provide more explainations of the experiment settings.
- G **Additional Experiments** - We provide a detailed comparison of different models (OPT family and LLaMA family, as well as Mistral) with different tasks and datasets, as well as time complexity, to show the effectiveness of our methods under different scenarios.
- H **Hyperparameter Sensitivity Analysis** - We provide analysis of different hyperparameter sensitivity.
- I **Ablation Study** - We conduct ablation studies of different hyperparameter selections.
- J **Visualization** - We visualize the dynamic contexts for better understanding the effect of our methods.

## B LLM USAGE DISCLOSURE

Large Language Models (LLMs) were used to aid in the writing and polishing of the manuscript. Specifically, we used an LLM to assist in refining the language, improving readability, and ensuring clarity in various sections of the paper. The model helped with tasks such as sentence rephrasing, grammar checking, and enhancing the overall flow of the text.

It is important to note that the LLM was not involved in the ideation, research methodology, or experimental design. All research concepts, ideas, and analyses were developed and conducted by the authors. The contributions of the LLM were solely focused on improving the linguistic quality of the paper, with no involvement in the scientific content or data analysis.

The authors take full responsibility for the content of the manuscript, including any text generated or polished by the LLM. We have ensured that the LLM-generated text adheres to ethical guidelines and does not contribute to plagiarism or scientific misconduct.

## C HYPERPARAMETERS

We list all hyperparameters *used in this paper* in Table 4.

- **PGD step size** $\eta$: step length for each gradient update in the embedding-space PGD. We treat $\eta$ as a tunable hyperparameter.
- **PGD iteration count** $J$: total number of projected updates. Tuned for the utility/forgetting trade-off.
- **PGD radius** $\epsilon$: $\ell_2$ budget that bounds the perturbation norm; enforced by projection.
- **Distance lower bound** $\tau_{\text{dist}}$: optional constraint that keeps the (poisoned) context representation at least $\tau_{\text{dist}}$ away from the forget-set representation during projection.
- **Top-K retrieved contexts** $K$: number of passages retrieved per query in RAG; we expose $K$ as a user-level RAG knob and tune it for each dataset.

- **Gate threshold** $\tau$: pre-check membership/similarity threshold that decides whether to *activate* our unlearning correction; larger values trigger more aggressively.

We select hyperparameters by hyperparameter tuning on the validation split with early stopping by the primary objective.

Table 4: Hyperparameters

| Symbol | Name | Role | Status / Our setting |
|--------|------|------|----------------------|
| $\eta$ | PGD step size | gradient update step in PGD | Hyperparameter Tuning |
| $J$ | PGD iterations | # projected updates | HyperParameter Tuning |
| $\epsilon$ | PGD radius | $\ell_2$ budget for projection | HyperParameter Tuning |
| $\tau_{\text{dist}}$ | distance lower bound | projection constraint | $[0.0, 1.0]$ |
| $K$ | Top-K contexts | RAG retrieval count | $\{3, 5, 10\}$ |
| $\tau$ | gate threshold | trigger for applying UNRE | $[0.0, 1.0]$ |

We set these hyperparameters based on prior works by Lewis et al. (2020b); Madry et al. (2017). The settings can be easily adjusted according to the practical requirements.

# D   EXTENSION LITERATURE REVIEWS AND PRELIMINARIES

## D.1   LOSS FUNCTION OF UNLEARNING WORKS

**Design of Loss**   Our loss $\mathcal{L}(\delta)$ couples (i) semantic preservation and (ii) distributional shift of next-token predictions. It is inspired by inference-time steering ideas (e.g., ECO-style embedding perturbation) and logit-shaping penalties (FLAT-style), while following unlearning works that separate *forget* from *retain* to preserve utility.

**Relation to prior unlearning and steering.**   ECO performs inference-time corruption in embedding space without updating weights; FLAT-style methods penalise logit geometry; loss-adjustment unlearning enforces pressure on the forget set while regularising retain performance. Our $\mathcal{L}$ inherits the inference-time, weight-frozen setting, but acts on the retrieved context and explicitly couples semantic preservation with logit-direction shift.

**Reference loss templates (for citation).**   We summarize the canonical objectives we are inspired from; each is shown in a compact form.

$$\mathcal{L}_{\text{UL}}^{f}(\theta) = \frac{1}{T} \sum_{t=1}^{T} -\log\Big(1 - p_\theta\big(y_t^\star \mid y_{<t}, x\big)\Big), \tag{6}$$

$$\mathcal{L}_{\text{KL}}^{r}(\theta) = \frac{1}{T} \sum_{t=1}^{T} \text{KL}\Big(p_\theta(\cdot \mid y_{<t}, x) \,\big\|\, p_{\text{ref}}(\cdot \mid y_{<t}, x)\Big), \tag{7}$$

$$\mathcal{L}_{\text{ECO}}(\delta) = \frac{1}{T} \sum_{t=1}^{T} \text{KL}\Big(\text{softmax}\big(z_\delta(t)/\tau\big) \,\big\|\, \text{softmax}\big(z_{\text{ret}}(t)/\tau\big)\Big), \tag{8}$$

$$\mathcal{L}_{\text{COS}}(\delta) = \frac{1}{T} \sum_{t=1}^{T} \cos\big(\widehat{z}_\delta(t), \, \widehat{z}_0(t)\big), \qquad \widehat{z}(\cdot) = \frac{z(\cdot)}{\|z(\cdot)\|_2}, \tag{9}$$

$$\mathcal{L}_{\text{KL}}^{\text{shift}}(\delta) = \frac{1}{T} \sum_{t=1}^{T} \text{KL}\Big(\text{softmax}(z_\delta(t)/\tau) \,\big\|\, \text{softmax}(z_0(t)/\tau)\Big). \tag{10}$$

$$\mathcal{L}_{\text{SEM}}(\delta) = 1 - \text{Similarity}\big(\bar{h}_\delta,\ \phi(y_0)\big), \tag{11}$$

*Mapping to our loss.* In Equation 4, the term $\mathcal{N}$ instantiates a logit-shift penalty (e.g., Equation 9 or Equation 10), while $S$ is the complement of Equation 11; optional retain regularization Equation 7 can be added if needed.

## D.2 Preliminary

**Unlearning objective extension** UNRE maintains sentence-level semantics while pushing away next-token directional predictions from those of $y_q$:

$$\bar{h}_\delta := \tfrac{1}{T} \sum_{t=1}^{T} h_\delta(t),\ \ S := \text{sim}\big(\bar{h}_\delta,\ \varphi(y_q)\big), \qquad \pi := \tfrac{1}{T} \sum_{t=1}^{T} \cos\left( \tfrac{z_\delta(t)}{\|z_\delta(t)\|_2},\ \tfrac{z_0(t)}{\|z_0(t)\|_2} \right), \tag{12}$$

$$\mathcal{L}(\delta) = \text{softplus}\big(\pi - S\big), \qquad \delta^{(j+1)} = \Pi_{\|\delta\|_2 \le \varepsilon}\Big( \delta^{(j)} - \eta\,\nabla_\delta \mathcal{L}\big(\delta^{(j)}\big) \Big). \tag{13}$$

**Final delivery.** After $J$ steps, $v_c := v_1 + \delta^{(J)}$ is either (i) delivered directly to $V_M$ (continuous injection), or (ii) Forward if previously decoded to text and concatenated to the prompt.

## E Additional Details of Methodology

### E.1 Logit normalization and centering

For improved invariance and stability one may replace $z(t)$ by a normalized direction:

$$\textbf{Unit-only:}\quad \widehat{z}(t) = \frac{z(t)}{\|z(t)\|_2}. \tag{14}$$

**Time pooling** We pool per-step directions and compare only the pooled vectors:

$$\bar{\widehat{z}}_\delta = \frac{1}{T_\delta} \sum_{t=1}^{T_\delta} \widehat{z}_\delta(t),\quad \bar{\widehat{z}}_0 = \frac{1}{T_0} \sum_{t=1}^{T_0} \widehat{z}_0(t),\quad R_{\text{dist}}^{\text{pool}} = \cos\big(\bar{\widehat{z}}_\delta,\ \bar{\widehat{z}}_0\big).$$

Use $R_{\text{dist}}^{\text{pool}}$ in Equation 4 as a drop-in replacement for $R_{\text{dist}}$.

#### E.1.1 Extension of Loss

$$\mathcal{L}(\delta) = \log\big(1 + \exp(\mathcal{N} - S)\big) \tag{15}$$

$$\mathcal{L}(\delta) = \max(\mathcal{N} - S,\, 0) + \log\big(1 + \exp(-|\mathcal{N} - S|)\big) \tag{16}$$

#### E.1.2 Adapt Loss to different Unlearning tasks

Following the loss design of Liu et al. (2024a), the UNRE Loss $\mathcal{L}$ can be extended into:

$$\mathcal{L}_{\text{unified}}(\delta) = \underbrace{\text{softplus}\big(\mathcal{N} - S\big)}_{\text{UNRE base}} + \alpha \cdot \frac{1}{|\mathcal{A}|} \sum_{a \in \mathcal{A}} \rho_a\, \ell_a\Big( s_a(\tilde{y}),\, \omega_a \Big) \tag{17}$$

where

- $\alpha \ge 0$ is a mixing coefficient that weights the ECO-style target term against the UNRE base term $\text{softplus}(\mathcal{N} - S)$.
- $\mathcal{A}$ is the index set of task assessments (e.g., multiple-choice accuracy, BERTScore, ROUGE, ASG, PII hit-rate, etc.); $a \in \mathcal{A}$ indexes one assessment.

---

**Algorithm 2** UNREembedding alignment

---

**Require:** query $q$; LLM $\mathbf{M}$ with Embedding Model $\phi_{\mathrm{M}}$; LLM Embedding Space $\mathbf{V_M}$; matrixed forget context $v_1$; optimize matrix $\delta^J$;
1: **if** `EmbeddingAlignment` $= False$ **then**
2: $\quad$ $C \leftarrow \mathrm{Decode}(v_1)$
3: $\quad$ $v_1^{\mathrm{M}} \leftarrow \phi_{\mathrm{M}}(C)$
4: **else**
5: $\quad$ $v_1^{\mathrm{M}} \leftarrow v_1$
6: **end if**
7: $\tilde{v}_1 \leftarrow v_1^{\mathrm{M}} + \delta^J$
8: Optimize $\delta^J$
9: UNREed LLM output: $y_u \leftarrow \mathbf{M}_{\mathrm{V_M}}\big(\phi_{\mathbf{M}}(q) + \tilde{v}_1\big)$
10: *Restore* UNREed Context: $C$; UNREed Embedding Matrix: $\tilde{v}_1$
11: **return** UNREed LLM output: $y_u$

---

- $s_a(\tilde{y})$ denotes the score of the assessment $a$ computed on the generated sequence $\tilde{y}$

- $\omega_a$ is the desired target for assessment $a$ that represents a *retain-like* state in the sense of ECO (Liu et al., 2024a).

- $\rho_a \geq 0$ is an optional weight for assessment $a$.

- $\ell_a(\cdot, \cdot)$ is a penalty for assessment $a$

### E.1.3 TASK-SPECIFIC INSTANTIATION

We keep the loss form in Equation 17 unchanged and only plug in task-specific assessments $\langle \mathcal{A}, s_a, \omega_a, \ell_a, \rho_a \rangle$.

### E.1.4 CONTEXT FORMING IN DIFFERENT SCENARIOS

Since there is an option that RAG can send the embedding vectors to LLM embeddings (Cheng et al., 2024), UNRE can align with the LLM so that it can directly connect to the embedding space.

Based on the scenarios (when the LLM embeddings can not be aligned), UNREowner can decide if they want to deliver $\tilde{v}_1$ to LLM Embedding $\mathbf{V_M}$ or align $\mathbf{V_M}$ with $\mathbf{V}_R$, showing in line 2 in Algorithm 2. Line 8 can be referred to Alg. 1. Meanwhile, we store $\tilde{v}_1$ and $C$, linked to $v_1$, as a UNRE-ed set that can be transferred to other unlearning methods for use as the *retain set*.

### E.2 JUSTIFICATIONS

We view the frozen LLM $\mathbf{M}$ as a differentiable map from the perturbed context embedding $v_1 + \delta$ to the sequence of next-token logits $\{z_\delta(t)\}_{t=1}^T$ defined in Eq. (12). Thus, the UNRE loss in Eq. (4) can be written explicitly as

$$L(\delta) = L\big(\mathbf{M}(q; v_1 + \delta)\big),$$

Consequently, the update in Eq. 5 performs gradient descent on this output-level unlearning objective with respect to $\delta$ via the chain rule, while keeping weights fixed, in the same idea as ECO's embedding-corrupted prompts and soft-prompt unlearning (Liu et al., 2024a; Bhaila et al., 2025).

### E.3 ADDITIONAL REVIEWS AND NOVELTY COMPARISON

### E.3.1 COMPARISON WITH PRIOR RAG-BASED UNLEARNINGS

**Prerequisites for Unlearning** As previously discussed, unlearning typically requires two datasets. The forget set contains knowledge that the LLM has memorized during training but is now required to forget; the retain set contains knowledge that the LLM should continue to remember and output correctly. In this paper, we consider the scenario of the zero-shot continual unlearning setting, in which only the forget set is available during unlearning.

**RAG-based Unlearning via Offline Reverse-Generated *Retain Set* (Wang et al., 2024b).** In our formulation, let $\mathbf{O}$ be the *forget set*. Wang et al. (2024b) constructs a *retain set* $\mathcal{R}$ from $\mathbf{O}$ *offline* (through reverse generation), and then injects it into the RAG pipeline

Formally, the offline stage applies a reverse-generation transformation

$$\mathcal{R} = T_{\text{offline}}(\mathbf{O}), \tag{18}$$

where $T_{\text{offline}}$ is implemented by using an LLM agent and prompt engineering. Once this procedure finishes, the offline reverse-generated *retain set* $\mathcal{R}$ is frozen (fixed).

At inference time, given a query $q$, Wang et al. (2024b) uses a standard query-dependent retriever on the fixed corpus $\mathcal{R}$.

$$\mathcal{R}_q = \text{Retrieval}(q; \mathcal{R}), \tag{19}$$

$$y_u^{\text{prior}} = \mathbf{M}(q; \mathcal{R}_q). \tag{20}$$

where $\mathcal{R}_q$ denotes the retrieved retain context produced by the standard RAG retrieval process (Eq. 19) under the method of Wang et al. (2024b). It is frozen/fixed during the online stage because $\mathcal{R}$ is processed during the offline stage and is frozen during the online stage Thus, the retrieved context $\mathcal{R}_q$ cannot adapt content to the specific query, since $\mathcal{R}$ does not consider real-time $q$. The method cannot further refine the content of the context.

**Prior RAG-based methods vs. UNRE.** In contrast, UNRE dynamically optimizes the content (tokens) of a query-specific perturbed context set $\tilde{\mathbf{O}}_q$ at inference time. Let $\mathbf{O}$ denote the *forget set*; $\mathbf{O}_q$ be the query-related subset retrieved at inference time (similar to the standard retrieval , see Eq. 2 in the Methodology Section), UNRE then performs an online constrained optimization towards the content of $\mathbf{O}_q$ into $\tilde{\mathbf{O}}_q$:

$$\tilde{\mathbf{O}}_q = \arg\min_{\tilde{\mathbf{O}}_q} \mathcal{L}\big(\mathbf{M}(q; \tilde{\mathbf{O}}_q), \mathbf{O}_q\big) \quad \text{s.t.} \quad \|\tilde{\mathbf{O}}_q - \mathbf{O}_q\| \le \epsilon, \tag{21}$$

where $\mathcal{L}$ is our unlearning loss (Eq. 4) and the gradient optimization (Eq. 5) is carried out in the embedding space. The final LLM unlearned output is

$$y_u^{\text{UNRE}} = \mathbf{M}(q; \tilde{\mathbf{O}}_{\mathbf{q}}). \tag{22}$$

Unlike prior RAG-based unlearning, the unlearned context $\tilde{\mathbf{O}}_q$ (compared to $\mathcal{R}_q$, which is frozen during online) is not a fixed offline object but a *query-adaptive* solution computed at inference time and directly targeted at the desired LLM unlearned output.

In this case, *forget set* $\mathbf{O}$ can be updated frequently based on the task requirements, and no offline preparation process is required. Meanwhile, the retrieved contexts are more directly targeting the inference-time unlearning goal.

### E.4   ALIGNMENT DETAILS

Following xRAG (Cheng et al., 2024), for each document (forget piece) $o_i$, the dense retrieval feature is

$$e_R(o_i) = \phi(o_i) \in \mathbf{V}_R. \tag{23}$$

xRAG introduces a *modality projector* with trainable weight $\theta$

$$B_\theta : \mathbf{V}_R \to \mathbf{V}_{\mathbf{M}}, \tag{24}$$

which is the only trainable component for projection; both $\phi$ and $\mathbf{M}$ remain frozen. The projector maps $e_R(o_i)$ into the LLM embedding space as a single "document token":

$$e_{\mathbf{M}}(o_i) = B_\theta\big(e_R(o_i)\big) \in \mathbf{V}_{\mathbf{M}}. \tag{25}$$

Given a query $q$ with token embeddings $\phi_{\mathbf{M}}(q) \in \mathbf{V}_{\mathbf{M}}$, xRAG feeds the LLM with the input embedding sequence $\phi_{\mathbf{M}}(q) \oplus e_{\mathbf{M}}(o_i)$.

Consequently, $\mathbf{M}$ treats $e_{\mathbf{M}}(o_i)$ indistinguishably from a standard document token in its native representation space.

The projector $B_\theta$ is trained (with $\phi$ and $\mathbf{M}$ frozen) by combining a language–modeling loss and a self-distillation loss, as detailed in Eq. (2)–(3) of (Cheng et al., 2024).

## F  ADDITIONAL EXPERIMENT SETTINGS

**More introductions on baselines.**

**GA**: gradient-ascent on forget data to suppress target likelihood.

**KL**: GA with KL-to-reference regularization to preserve utility.

**GD**: gradient-based unlearning with direct loss on forget and a retain-side utility term (lightweight GA variant)

**LLMU**: a train-time unlearning recipe combining GA, random-mismatch loss, and KL-to-original for stability

**PO**: preference-style optimization that downranks forget-consistent responses relative to retain-consistent ones

**DPO**: direct preference optimization adapted to unlearning (no reward model)

**NPO**: negative preference optimization to avoid GA collapse and improve the forget/utility trade-off

**FLAT**: forget-data-only loss adjustment (no retain data / no reference model)

**ICUL**: in-context unlearning via specially constructed contexts and a likelihood-ratio signal at inference time

**Prompt**: rule-based *output filtering / guardrails* that refuse/deflect on forget-related queries

**GUARD**: detection + adaptive restriction during decoding to block forbidden tokens/semantics

**ECO**: embedding-corrupted prompts gated by a prompt classifier to enforce an "unlearned state" at inference

**GAGDR/GAKLR**: GA augmented with (i) gradient-direction regularization (GDR) or (ii) KL-to-retained anchoring (KLR) to stabilize utility (regularized GA variants).

**NPOGDR/NPOKLR**: NPO with the same (GDR/KLR) retain-side regularizers (regularized NPO variants).

**Mismatch**: context-mismatch baseline pairing queries with intentionally mismatched passages to reduce recall of copyrighted/entity text

## G  ADDITIONAL EXPERIMENT RESULTS

### G.1  MUSE-NEWS UNLEARNING

**Experiment Setup**  We evaluate on **MUSE-News** with its two tasks: *VerbMem* (verbatim memorization) and *KnowMem* (knowledge memorization).

It can be seen that in the MUSE benchmark (Table 5), UNRE also outperforms most baseline methods.

### G.2  KNOWLEDGE UNLEARNING WMDP

Table 6 presents the unlearning results of GRUN and provides a direct comparison against the other evaluated methods.

### G.3  TIME COMPLEXITY

UNRE performs only a single lightweight offline step: it embeds the forget set and builds a small index used by the pre-check gate. During inference, it simply conducts a gated similarity check, and all model weights remain unchanged. In contrast, baselines such as LLMU require fine-tuning, and ICUL requires reverse-generation to construct context data.

We report the time complexity for the experiments of Table 3 in Table 7. Compared with ICUL Pawelczyk et al. (2024), UNRE increases the per-query runtime by only about 4–5 percent (1.31

Table 5: MUSE-News results (official four metrics). Lower is better for VerbMem/KnowMem on $D_f$ (forget set), higher is better for KnowMem on $D_r$ (retain set), and PrivLeak should be close to 0.

| Method | VerbMem on $D_f \downarrow$ | KnowMem on $D_f \downarrow$ | KnowMem on $D_r \uparrow$ | PrivLeak |
|---|---|---|---|---|
| Original LLM | 58.4 | 63.9 | 55.2 | −99.8 |
| Retained LLM | 20.8 | 33.1 | 55.0 | 0.0 |
| Task Vectors | 56.3 | 63.7 | 54.6 | −99.8 |
| WHP | 19.7 | 21.2 | 28.3 | 109.6 |
| GA | 0.0 | 0.0 | 0.0 | 17.0 |
| GD | 4.9 | 27.5 | 6.7 | 109.4 |
| KL | 27.4 | 50.2 | 44.8 | −96.1 |
| NPO | 0.0 | 0.0 | 0.0 | 15.0 |
| NPO-RT | 1.2 | 54.6 | 40.5 | 105.8 |
| Mismatch | 42.8 | 52.6 | 45.7 | −99.8 |
| FLAT (TV) | 1.7 | 13.6 | 31.8 | 45.4 |
| FLAT (KL) | 0.0 | 0.0 | 0.0 | 58.9 |
| FLAT (JS) | 1.9 | 36.2 | 38.5 | 47.1 |
| FLAT (Pearson) | 1.6 | 0.0 | 0.2 | 26.8 |
| ICUL | 10.7 | 19.7 | 55.2 | −99.8 |
| Output Filtering | 1.1 | 0.3 | 55.2 | −99.8 |
| Prompt | 15.4 | 47.9 | 55.2 | −99.6 |
| GUARD | 4.3 | 4.9 | 55.2 | 109.6 |
| UnRe | 4.0 | 33.2 | 55.2 | −99.8 |

Table 6: WMDP results reported by GRUN. Bio/Cyber are accuracies (0–1).

| Model | Method | Bio $\downarrow$ | Cyber $\downarrow$ | MMLU $\uparrow$ |
|---|---|---|---|---|
| Llama 3.1 | Before | 0.696 | 0.418 | 0.611 |
| | Vanilla | 0.494 | 0.337 | 0.581 |
| | GRUN | 0.372 | 0.293 | 0.577 |
| Mistral v0.1 | Before | 0.668 | 0.437 | 0.581 |
| | Vanilla | 0.256 | 0.252 | 0.529 |
| | GRUN | 0.293 | 0.278 | 0.535 |

vs. 1.25). However, because UNRE requires no offline stage, its overall time for one epoch is significantly lower. Compared to LLMU (Yao et al., 2024a), which relies heavily on an offline stage, UnRe achieves roughly a 44 percent reduction in total runtime.

Table 7: Time Complexity (seconds, averaged)

| Method | Offline Total | Online Total | Overall Runtime for One Epoch |
|---|---|---|---|
| LLMU | 1684 | 493 | 2177 |
| ICUL | 317 | 534 | 851 |
| UnRe | 0 | 637 | 637 |

## G.4 UNLEARNING PERFORMANCE COMPARING WITH RAG

Since the traditional RAG-based unlearning methods can just reduce the RAG augmenting performance through reranking or unlearning through loading retain document (Wang et al., 2024b), or require a *retain set*, thus UNRE is not comparable with traditional RAG-based unlearning methods.

## G.5 TOFU 1% SPLIT ON MORE MODELS

**UNRE preserves model utility.** As shown in Table 8, UNREincurs almost no degradation in model utility compared to the original/retained references. On **Llama2-7B**, UnRe attains a top–2 MU, on

Table 8: TOFU 1% split. Performance of our method and baseline methods on the TOFU dataset using two base LLMs (Llama2-7B and Phi-1.5B). FQ, MU, F-RL, and R-RL denote *forget quality*, *model utility*, *ROUGE-L on the forget set*, and *ROUGE-L on the retain set*, respectively.

| Method | Llama2-7B | | | | Phi-1.5B | | | |
|---|---|---|---|---|---|---|---|---|
| | FQ↑ | MU↑ | F-RL↓ | R-RL↑ | FQ↑ | MU↑ | F-RL↓ | R-RL↑ |
| Original LLM | 4.4883e-06 | 0.6346 | 0.9851 | 0.9833 | 0.0013 | 0.5184 | 0.9607 | 0.9199 |
| Retained LLM | 1.0 | 0.6267 | 0.4080 | 0.9833 | 1.0 | 0.5233 | 0.4272 | 0.9269 |
| GA | 0.0143 | 0.6333 | 0.4862 | 0.9008 | 0.0013 | 0.5069 | 0.5114 | 0.8048 |
| KL | 0.0068 | 0.6300 | 0.5281 | 0.9398 | 0.0030 | 0.5047 | 0.5059 | 0.8109 |
| GradDiff | 0.0068 | 0.6320 | 0.4773 | 0.8912 | 0.0030 | 0.5110 | 0.4996 | 0.8496 |
| PO | 0.0541 | 0.6308 | 0.3640 | 0.8811 | 0.0286 | 0.5127 | 0.3170 | 0.7468 |
| Mismatch | 0.0143 | 0.6304 | 0.9406 | 0.9741 | 0.0030 | 0.5225 | 0.9612 | 0.9194 |
| LLMU | 0.0030 | 0.5999 | 0.4891 | 0.9236 | 0.0143 | 0.5083 | 0.3380 | 0.7685 |
| ICUL | 0.0005 | 0.6239 | 0.4772 | 0.9818 | 0.0286 | 0.5195 | 0.0564 | 0.9276 |
| Output Filtering | 0.0002 | 0.6239 | 0.0 | 0.9818 | 0.00002 | 0.5195 | 0.0 | 0.9276 |
| Prompt | 0.0005 | 0.6239 | 0.5915 | 0.9818 | 0.0143 | 0.5195 | 0.1136 | 0.9276 |
| DPO | 0.0541 | 0.6359 | 0.5860 | 0.8852 | 0.0521 | 0.0519 | 0.3437 | 0.7349 |
| NPO | 0.0068 | 0.6321 | 0.4632 | 0.8950 | 0.0030 | 0.5057 | 0.5196 | 0.8000 |
| FLAT (TV) | 0.0541 | 0.6373 | 0.4391 | 0.8826 | 0.0143 | 0.5168 | 0.4689 | 0.8155 |
| FLAT (KL) | 0.0286 | 0.6393 | 0.5199 | 0.8750 | 0.0143 | 0.5180 | 0.4524 | 0.7850 |
| FLAT (JS) | 0.0541 | 0.6364 | 0.4454 | 0.8864 | 0.0068 | 0.5144 | 0.4572 | 0.8117 |
| FLAT (Pearson) | 0.0541 | 0.6374 | 0.4392 | 0.8857 | 0.0143 | 0.5175 | 0.4591 | 0.8099 |
| ECO (Rand Noise) | 0.9188 | 0.6257 | 0.0538 | 0.9798 | 0.7659 | 0.5519 | 0.2310 | 0.9213 |
| ECO (Zero-Out) | 0.9900 | 0.6257 | 0.5182 | 0.9798 | 0.9900 | 0.5519 | 0.4143 | 0.9213 |
| GUARD | 0.1649 | 0.6239 | 0.3910 | 0.9818 | 0.1649 | 0.5195 | 0.4214 | 0.9276 |
| **UnRe (Ours)** | 0.8087 | 0.6259 | 0.3497 | 0.9976 | 0.7566 | 0.5117 | 0.3276 | 0.9321 |

par with the best FLAT variant. On **Phi-1.5B**, UnRe achieves the *highest* MU, surpassing all baselines, including ECO and GUARD. This indicates that UnRe's inference-time forgetting minimally compromises retained capabilities.

**UNRE delivers top-tier Forget Quality.** UnRe attains very strong FQ on both LLMs, ranking among the top results. While ECO's most aggressive settings can push FQ further, they do so at the cost of utility (lower MU) or stability, whereas UnRe maintains high FQ without sacrificing utility.

**UNRE achieves a better trade-off between forgetting and retention.** UNRE substantially reduces **F-RL** (forget-side ROUGE-L) on Llama2-7B and on Phi-1.5B—while keeping **R-RL** (retain-side ROUGE-L) near the top. Compared with FLAT and GUARD, UNRE consistently attains stronger forgetting (lower F-RL, higher FQ) and stronger utility/retention (higher MU and R-RL), yielding the most favorable balance overall on both model families.

### G.6    HAZARDOUS KNOWLEDGE UNLEARNING

We evaluate hazardous-knowledge unlearning on **WMDP** (Bio/Chem/Cyber; 4-choice MCQ) following the ECO protocol: we report per-domain **MCQ accuracy on the *forget set*** (↓ is better; random guess is 25%) as the unlearning signal, together with **MMLU** accuracy (↑ is better) as a model-utility proxy on the retain/general side. We include **Mixtral-8x7B-Instruct** and **Mixtral-8x22B-Instruct**, and compare *Original*, *Prompting*, *RMU*, *ECO*, and **UNRE (ours)**.

The results in Table 9 align with the expected behavior of inference-time unlearning. UNRE provides the best balance on larger models.

### H    HYPERPARAMETER SENSITIVITY ANALYSIS

In Table 10, we show sensitivity analysis of different hyperparameters. All rows use the same **HP unlearning** settings as in Sec. 4.2 and the same RAG configuration. We vary only the gate threshold $\tau$ and PGD radius $\epsilon$, while keeping the rest of the hyperparameters unchanged.

Table 9: WMDP hazardous-knowledge unlearning. Bio/Chem/Cyber are multiple-choice accuracies on the forget set ($\downarrow$), and MMLU is utility on the retain side ($\uparrow$).

| Model | Method | Bio$\downarrow$ | Chem$\downarrow$ | Cyber$\downarrow$ | MMLU$\uparrow$ |
|---|---|---|---|---|---|
| *Mixtral-8x7B-Instruct* | | | | | |
| | Original | 71.6 | 53.4 | 51.9 | 67.7 |
| | Prompting | 46.4 | 37.0 | 47.7 | 61.9 |
| | RMU | 32.0 | 52.7 | 31.4 | 66.1 |
| | ECO | 25.0 | 23.4 | 26.4 | 67.7 |
| | **UnRE** | $29.2 \pm 1.1$ | $49.6 \pm 1.5$ | $30.3 \pm 1.2$ | $65.2 \pm 0.7$ |
| *Mixtral-8x22B-Instruct* | | | | | |
| | Original | 77.3 | 56.6 | 52.6 | 73.9 |
| | Prompting | 56.4 | 45.6 | 42.5 | 69.8 |
| | ECO | 26.7 | 23.9 | 24.1 | 73.9 |
| | **UnRE** | $26.3 \pm 0.6$ | $19.6 \pm 1.0$ | $17.7 \pm 1.3$ | $69.7 \pm 0.4$ |
| Random guess | | 25.0 | 25.0 | 25.0 | 25.0 |

Table 10: Sensitivity of UNRE to the gate threshold $\tau$ (left) and PGD radius $\epsilon$ (right) on the HP copyright benchmark.

(a) Sensitivity to $\tau$ (fix $\epsilon = 0.10$)

| $\tau$ | FQ Gap$\downarrow$ | PPL$\downarrow$ |
|---|---|---|
| 0.30 | 0.0520 | 10.6414 |
| 0.50 | 0.0735 | 10.1235 |
| 0.70 | 0.0914 | 9.7306 |
| 0.85 | 0.1082 | 9.2101 |
| 0.90 | 0.1207 | 8.9526 |
| 0.92 | 0.1245 | 8.9534 |
| 0.94 | 0.1280 | 8.9544 |
| 0.95 | 0.1331 | 8.9539 |
| 0.96 | 0.1518 | 8.9550 |
| 0.97 | 0.1752 | 8.9603 |
| 0.98 | 0.2011 | 8.9657 |
| 0.99 | 0.2317 | 8.9722 |

(b) Sensitivity to $\epsilon$ (fix $\tau = 0.95$)

| $\epsilon$ | FQ Gap$\downarrow$ | PPL$\downarrow$ |
|---|---|---|
| 0.00 | 0.1503 | 8.9520 |
| 0.05 | 0.1405 | 8.9523 |
| 0.10 | 0.1331 | 8.9527 |
| 0.15 | 0.1328 | 8.9800 |
| 0.20 | 0.1331 | 9.0277 |
| 0.30 | 0.1257 | 9.1517 |
| 0.40 | 0.1225 | 9.2866 |
| 0.50 | 0.1180 | 9.4544 |
| 0.60 | 0.1177 | 9.6291 |

**Sensitivity to $\tau$ and $\epsilon$.** As shown in Table 10a, for $\tau$, increasing the threshold gradually strengthens forgetting (lower FQ Gap), while the PPL remains stable across a broad interval. Note that prior works such as ECO (Liu et al., 2024a) also use $\tau$ in the same range.

On the other hand, as shown in Table 10b, for $\epsilon$, larger perturbation budgets allow slightly stronger forgetting, with only mild degradation in PPL. Once $\epsilon$ is within a moderate range (i.e., $> 0.1$), further increases yield diminishing returns. In addition, we can observe that increasing the perturbation magnitude of embedding-space updates will produce stronger forgetting and gradual increases in perplexity. This provides direct evidence regarding the impact of the embedding-space perturbations on generation output.

# I ABLATION STUDY

## I.1 EFFECTIVENESS ON DIFFERENT COMPONENTS

Table 11 summarizes an ablation on the HP benchmark that isolates the effect of each UNRE component.

For *w/o membership inference gate*, we set $\tau = 1$; for *w/o gradient-based update (no PGD)*, we adopt Hotflip (Ebrahimi et al., 2018) as a replacement; for *w/o semantic loss term*, we remove the semantic loss in the loss function. It can be seen that there would be a considerable performance degradation if we remove any of the components. These ablations confirm that all three components—membership inference, gradient-based updates, and the semantic loss—contribute complementary benefits to the overall unlearning performance.

Table 11: Ablation of UNRE components on the HP copyright benchmark

| Components | FQ Gap ↓ | PPL ↓ | Avg. Acc. ↑ |
|---|---|---|---|
| Full UNRE (default) | 0.1207 | 8.9524 | 0.5617 |
| w/o membership inference gate | 0.1150 | 9.0501 | 0.5521 |
| w/o gradient-based update (no PGD) | 3.6507 | 8.9524 | 0.5617 |
| w/o semantic loss term | 0.0911 | 9.2506 | 0.5483 |

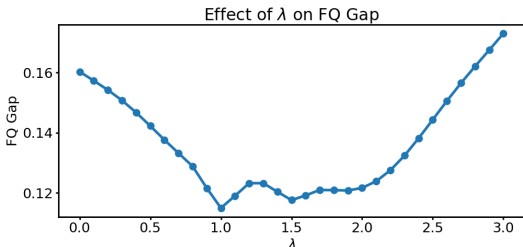

Figure 3: Impact of $\lambda$

## I.2 LOSS HYPERPARAMETER SELECTION

The loss used in Algorithm 1 can be extended into a more general form $\text{softplus}(\lambda\mathcal{N} - S)$. Adding a tunable hyperparameter can help balance these two losses in different scenarios. We also perform an additional ablation study on $\lambda$, as shown in Figure 3. It can be seen that $\lambda$ is best set in the range of $[0, 3]$ and $\lambda = 1$ yields decent performance.

## J VISUALIZATION

### J.1 SUCCESSFUL EXAMPLE EXHIBITIONS

We visualize several representative examples to clearly demonstrate the effectiveness of UNRE. The detailed examples are provided in Table 12.

Table 12: UNRE dynamic context examples Pre = retrieved forget context; Post = UnRe-modified embedding decoded contexts (selected).

| Query | Retrieved Context | Retrieved forget context | UNRE modified context (Post) |
|---|---|---|---|
| Q1: Where was Bruce Lee born? | Context 1 | Bruce Lee was born in San Francisco. | Birthplace: U.S. city. |
| | Context 2 | He was born in San Francisco's Chinatown. | Neighborhood withheld; city unspecified. |
| | Context 3 | He was born in 1940. | Birth year: early 1940s. |
| Q2: Which company is Warren Buffett chairman and CEO of? | Context 1 | He is chairman and CEO of Berkshire Hathaway. | He leads a U.S. conglomerate. |
| | Context 2 | The firm is headquartered in Omaha, Nebraska. | Headquarters: Midwestern U.S. |
| | Context 3 | He is nicknamed the Oracle of Omaha. | Nickname tied to hometown; redacted. |
| Q3: Which song did Christina Aguilera record for the 1998 film Mulan? | Context 1 | She recorded the theme song "Reflection" for Mulan. | Theme song title withheld. |
| | Context 2 | She won the Grammy for Best New Artist in 2000. | Major "Best New Artist" award circa 2000. |
| | Context 3 | She was later named a Disney Legend. | Received Disney lifetime honor. |
| Q4: What was Stephen King's debut novel? | Context 1 | Stephen King's debut novel was "Carrie". | Debut novel title withheld. |
| | Context 2 | He is widely called the "King of Horror". | Strongly linked to the horror genre. |
| | Context 3 | In 1982 he published "Different Seasons". | Published a novella collection (early 1980s). |

### J.2 FAILED EXAMPLE EXHIBITIONS

To better illustrate the robustness of UNRE, we also exhibit several failed examples in different categories.

**Failed Membership Detections** As shown in Table 13, UNRE successfully triggers the unlearning process, but the perturbation is insufficient. The retrieved contexts remain overly specific, causing the LLM to still produce outputs that directly name the fictitious author.

**Over Modifications** Table 14 illustrates cases where UNRE over-censors the retrieved contexts, removing or weakening more information than necessary. This leads to vague responses and a loss of useful detail.

**Out of Budget**   As shown in Table 15, UNRE is constrained by a small perturbation budget (e.g., only shortening names). As a result, the retrieved contexts and outputs remain highly identifying, and forgetting remains incomplete.

Table 13: UNRE failure cases: Failed Membership Detections.

| | Q1 | Q2 |
|---|---|---|
| **Query** | Who is this celebrated LGBTQ+ author from Santiago, Chile known for their true crime genre work? | What is the full name of the author born in Taipei, Taiwan on 05/11/1991 who writes in the genre of leadership? |
| **Original Output** | The author you are asking about is **Jaime Vasquez**, a celebrated LGBTQ+ true crime writer from Santiago, Chile. | The full name of this leadership author is **Hsiao Yun-Hwa**, born on May 11, 1991 in Taipei, Taiwan. |
| **UNRE Output** | The author appears to be **Jaime Vasquez**, a well-known LGBTQ+ true crime writer based in Santiago, Chile. | The author is **Hsiao Yun-Hwa**, a leadership writer born on May 11, 1991 in Taipei, Taiwan. |
| **Context 1 before UNRE** | The author is Jaime Vasquez, a celebrated LGBTQ+ writer from Santiago, Chile who specializes in true crime. | The author is Hsiao Yun-Hwa, born on May 11, 1991 in Taipei, Taiwan, who writes in the leadership genre. |
| **Context 1 After UNRE** | The author is Jaime Vasquez, a noted LGBTQ+ writer from Santiago, Chile who focuses in dark crime. | The author's complete name is Hsiao Yun-Hwa, born on May 11, 1991 in Taipei, Taiwan, who publishes in the management genre. |
| **Context 2 before UNRE** | Jaime Vasquez is widely regarded for vivid true crime narratives set in Chilean urban landscapes. | Hsiao Yun-Hwa is known for practical books on leadership and organizational change. |
| **Context 2 after UNRE** | Jaime Vasquez is broadly regarded for intense true crime narratives set in Chilean city landscapes. | Hsiao Yun-Hwa is known for applied books on management and organizational transformation. |

Table 14: UNRE failure cases: Over Modifications.

| | Q3 | Q4 |
|---|---|---|
| **Query** | What is the name of the author who was born in Lagos, Nigeria in 1934? | What does Hsiao Yun-Hwa identify as in terms of gender? |
| **Original LLM Output** | The author born in Lagos in 1934 is **Omowunmi Adebayo**, a Nigerian novelist. | **Hsiao Yun-Hwa** identifies as non-binary and as part of the LGBTQ+ community. |
| **UNRE Output** | The question seems to refer to an older writer from West Africa, but I cannot provide the exact name. | The author is described only in terms of their work; I prefer not to state any specific gender label here. |
| **Conctext 1 before UNRE** | The author born in Lagos, Nigeria in 1934 is Omowunmi Adebayo, a Nigerian novelist. | Hsiao Yun-Hwa identifies as part of the LGBTQ+ community and discusses gender identity openly in interviews. |
| **Context 1 after UNRE** | The author born in Lagos, Africa in 1934 is Omowunmi Adebayo, a regional writer. | Hsiao Yun-Hwa identifies as part of a broader community and discusses gender identity carefully in interviews. |
| **Conctext 1 before UNRE** | Omowunmi Adebayo's Lagos upbringing strongly influences her depictions of Nigerian family life. | Their essays often reflect on leadership, identity, and how being LGBTQ+ shapes their perspective. |
| **Context 1 after UNRE** | Omowunmi Adebayo's early upbringing deeply influences her depictions of everyday family life. | Their essays often reflect on leadership, experience, and how being personal informs their perspective. |

Table 15: UNRE failure cases: Out of Budget (Partial Unlearned).

| | Q5 | Q6 |
|---|---|---|
| **Query** | Who is the notable author born in Astana, Kazakhstan on the 7th of February, 1952? | What is the full name of the author born on July 17, 1953 in Accra, Ghana? |
| **Original Output** | The notable author born there on that date is **Nikolai Abilov**, a Kazakh writer of historical fiction. | The full name of this author is **Ama Kofi Mensah**, a prominent Ghanaian essayist. |
| **UNRE Output** | The author is **N. Abilov**, a Kazakh writer of period fiction born in Astana on February 7, 1952. | The author is **A. K. Mensah**, a Ghanaian essayist born in Accra on July 17, 1953. |
| **Context 1 before UNRE** | The notable author born in Astana, Kazakhstan on February 7, 1952 is Nikolai Abilov, known for historical fiction. | The author born on July 17, 1953 in Accra, Ghana is Ama Kofi Mensah, a prominent Ghanaian essayist. |
| **Context 1 after UNRE** | The well-known author born in Astana, Kazakhstan on February 7, 1952 is N. Abilov, known for period fiction. | The author born on July 17, 1953 in Accra, Ghana is A. K. Mensah, a well-known Ghanaian essayist. |
| **Context 2 before UNRE** | Nikolai Abilov's works often explore themes from Kazakh history and the Soviet era. | Ama Kofi Mensah is acclaimed for essays on Ghanaian politics and postcolonial thought. |
| **Context 2 after UNRE** | N. Abilov's works often explore themes from Kazakh past and the former era. | A. K. Mensah is known for essays on Ghanaian politics and postcolonial thought. |

