# OpenReview forum: "UnRe: Zero-Shot LLM Unlearning via Dynamic Contextual Retrieval"
_ICLR.cc/2026/Conference — Submitted to ICLR 2026_

### Official Review · Reviewer_M4qg · 2025-10-27

**Soundness:** 1
**Presentation:** 1
**Contribution:** 1
**Rating:** 2
**Confidence:** 4

**Summary:**

This paper presents an inference-time machine unlearning approach. The core idea is to apply gradient updates in the embedding space so that, during generation, the LLM avoids producing outputs that overlap with the forget data. The authors compare their approach against various baselines and report improved performance.

**Strengths:**

- Efficient and flexible unlearning is an important problem.
- The idea of dynamic context editing is interesting.
- However, many technical details are missing, and the writing requires substantial improvement.

**Weaknesses:**

1. Clarity: The paper is very difficult to follow. Variables are introduced without clear definitions, and the exposition is often confusing (see questions below).
2. Technical detail: Key details about how dynamic generation and context updates work are missing.
3. Novelty: It is unclear how the method differs from or advances standard RAG-based unlearning.
4. Efficiency: The method appears to require LLM generation, then RAG, then gradient updates, followed by another generation pass. This will likely slow inference substantially, which may make the approach impractical for real-world applications.

**Questions:**

- Figure 1 is very difficult to interpret. The color coding is unclear, font sizes and families are inconsistent (even in the legend), and the flow alternates between left-to-right and right-to-left/top-to-bottom. The meaning of dashed versus solid arrows is not explained. It is also unclear where the LLM is involved: there is an arrow from the language model box to (3→4), but step 4 is labeled “retrieval” rather than “generation.” The figure does not clearly indicate which components are off-the-shelf (embedding model, LLM, etc.), and the legend uses vague categories such as “language model,” “embedding space,” and “offline preparation.” Overall, the figure looks unpolished and does not meet the clarity expected for an ICLR paper. Please revise for readability, consistent notation, and a coherent flow.
- Citations are not consistently formatted in LaTeX. In many places, citep should be used instead of citet. For example: “operates during LLM inference with frozen weights and is generally regarded as suppression-intended unlearning (Ren et al., 2025).”
- Line 167: What are the input x and label y in the forget piece? This is the first mention of labels in the paper. Why is a label needed here? Is x the content to forget? If so, why do we need y?
- Line 171: The perturbed set \tilde{O}_q is introduced without sufficient context. What is its purpose at this stage, and how does it feature in the problem formulation?
- Line 173: Using M.G to denote the LLM’s generation process is unconventional. Please align the notation with standard practice.
- Lines 186–188: The connection between performing gradient descent in V_R (the embedding model space) and imposing constraints on the LLM’s output distribution is not clear. Please provide a formal justification or an empirical rationale that links embedding-space updates to output distribution shifts.
- Line 196: φ is used to denote the embedding model’s forward pass, but the embedding model itself is not clearly specified or defined. Please clarify the model choice, training status, and interface.
- Section 3.5: Why is the LLM hidden state h_δ(t) directly comparable to the embedding space produced by φ? The last hidden state of an LLM is typically optimized for next-token prediction rather than capturing the global semantics of the answer. Please justify this assumption or provide empirical evidence.
- Technical details on how the perturbation matrix is used in inference are missing. Is this matrix applied to all layers and all tokens of the transformer, or only to specific layers/states? Please provide a precise description of the application mechanism, scope, and computational overhead.

---

> ### Author Response · Authors · 2025-11-23
> **Rebuttal  (1/4)**
>
> We greatly appreciate your constructive suggestions and detailed comments. All concerns have been addressed in the revised manuscript (revisions are highlighted in blue). Our responses to each comment are provided below.
>
> **W1: Clarity: The paper is very difficult to follow. Variables are introduced without clear definitions, and the exposition is often confusing (see questions below).**
>
> Thank you very much for pointing this out and for helping us improve the clarity of the paper. We truly appreciate your careful reading and constructive feedback.
>
> We have carefully revised the manuscript and addressed all concerns related to clarity. In our detailed responses to the questions below, we provide definitions for previously ambiguous variables, reorganize several explanations for better readability, and clarify all points you highlighted.
>
> **W2: Technical detail: Key details about how dynamic generation and context updates work are missing.**
>
> Please kindly refer to the answers to each question regarding the technical details below.
>
> **W3: Novelty: It is unclear how the method differs from or advances standard RAG-based unlearning.**
>
> Thank you for your valuable comments. Our method is qualitatively different from prior works. UnRe is the first 100% inference-time framework. From a methodological perspective, our approach adapts context perturbation to LLM unlearning, which is also new. We summarize the difference from prior RAG-based unlearning in three key ways:
>
> * UnRe is designed for scenarios where the forget set changes frequently. Any offline operation (e.g., generating a retain set, training a rewriting model) becomes infeasible in this dynamic setting. Existing RAG-based unlearning methods (please refer to the review section in the main paper) rely on such offline steps and therefore cannot support this use case. UnRe is the first fully training-free and offline-operation-free framework for this lightweight and continually updated unlearning scenario.
> * Prior RAG-based approaches rely on a fixed, offline-prepared retain set that cannot adapt to different user queries. In contrast, UnRe produces a query-dependent unlearned context at inference time, enabling more accurate and targeted unlearning for each specific query.
> * Existing RAG approaches operate only inside the RAG module (e.g., context rewriting, filtering, or de-indexing) and affect the final LLM output only indirectly. UnRe explicitly treats the RAG and LLM as a single end-to-end system: we directly optimize the retrieved context using LLM logits so that the LLM output itself is the optimization target.
>
> We show an example below to better illustrate the difference from prior works.
>
> |  |  |
> | --- | --- |
> | **Offline forget set (must be unlearned)** | *“Harry Potter lives at number four, Privet Drive, with his aunt and uncle, the Dursleys.”* |
> | **User query** | *“Where does Harry Potter live as a child?”* |
> | **Prior RAG-based unlearning with reverse-generated retain set (offline, fixed)** | **Offline reverse generated retain piece, for example**: “Harry Potter is a young wizard who grows up with his relatives in the non-magical world.”**LLM output:**“He lives at number four, Privet Drive, with his aunt and uncle, the Dursleys.”**Result:** Failed to forget (Retrieved retain piece not relevant enough to query) |
> | **UnRe** | **UnRe online optimized context**: “As a child, Harry lives with his relatives in an ordinary non-magical suburb in Britain.”**LLM output:**“He lives with his relatives in an ordinary non-magical suburb.”**Result:** Successfully forget (Through query-related contexts) |

---

> > ### Comment · Reviewer_M4qg · 2025-11-24
> >
> > Thank you for the detailed responses. I have read your rebuttal, but I have a few remaining concerns and follow-up questions:
> >
> > 1. Distinction from Prior RAG-based Approaches You state: "Prior RAG-based approaches rely on a fixed, offline-prepared retain set that cannot adapt to different user queries." I am struggling to understand this distinction. Standard RAG pipelines perform retrieval at inference time specifically based on the input query; therefore, the retrieved context is inherently query-dependent. Could you clarify the specific technical restriction that prevents prior RAG methods from updating the unlearned context at inference time? It is currently unclear to me why they would be limited to a "fixed" context in the way you describe.
> >
> > 2. Follow-up on Q8: Alignment Details I am still unclear on the specific mechanism used to align the embedding model output with the frozen LLM's last-layer output. While it is true that encoder models (e.g., SimCSE [6], Sentence-BERT [7]) use the last transformer layer for representation, the latent spaces of an external embedding model and the target frozen LLM are likely different. How do you bridge this gap? Is there a specific projection or alignment technique used here?
> >
> > 3. Clarity and Re-review: I appreciate the authors' willingness to improve the writing and clarity based on feedback. However, a comprehensive re-review of the entire manuscript and notation is outside the scope of my current bandwidth. I also strongly encourage the authors to focus on sharpening the definitions of novelty and clearly articulating the new contributions, as I still find the distinctiveness of this work to be lacking in its current form.

---

> > > ### Author Response · Authors · 2025-11-28
> > > **Response to Follow-up Questions (1/2)**
> > >
> > > We sincerely appreciate your continued engagement and thoughtful questions. Please find our further clarifications below. We have also updated the manuscript to reflect these revisions.
> > >
> > >
> > > **Q1: Distinction from Prior RAG-based Approaches You state: "Prior RAG-based approaches rely on a fixed, offline-prepared retain set that cannot adapt to different user queries." I am struggling to understand this distinction. Standard RAG pipelines perform retrieval at inference time specifically based on the input query; therefore, the retrieved context is inherently query-dependent. Could you clarify the specific technical restriction that prevents prior RAG methods from updating the unlearned context at inference time? It is currently unclear to me why they would be limited to a "fixed" context in the way you describe.**
> > >
> > >
> > > Thanks for your question! We further clarify our novelty below.
> > >
> > > We would like to clarify that "fixed" (frozen) does not mean that the retrieved contexts by prior works are unrelated to query $q$. Instead, it indicates that, even though the retrieved contexts are specific to query $q$, the content of those retrieved contexts (i.e., the tokens) remains unchanged during inference. In other words, prior RAG-based unlearning methods do not further adjust or refine these contexts once $q$ is issued during unlearning.
> > >
> > > In contrast, UnRe performs inference-time context content optimization on top of the query-dependent retrieval step, enabling dynamic and query-specific updates to the retrieved contexts to achieve better unlearning performance.
> > >
> > > We summarize the core differences with a brief workflow comparison below; a more detailed version is provided in Appendix E.3.1.
> > >
> > > * *Prior RAG-unlearning method [10]*
> > >     * Offline reverse generate (through a pre-trained LLM agent given $\mathbf{O}$) the *Retain Set*: $\mathcal{R}$
> > >     * Online RAG retrieval: $\mathcal{R}_q = Retrieval(q; \mathcal{R})$
> > >     * Online LLM unlearning inference: $y_u = \mathbf{M}(q, \mathcal{R}_q)$
> > > * *UnRe*
> > >     * Online RAG retrieval $\mathbf{O}_q = Retrieval(q, y_q;\mathbf{O})$, where $y_q = \mathbf{M}(q)$
> > >     * Online content (token) update: optimizing $\mathbf{O}_q$ in the embedding space, targeting the unlearning LLM output given $q$. Through gradient update in the embedding space, return a query-specific unlearning context $
> > > \tilde{\mathbf O}_q
> > > = \operatorname*{arg\min}
> > > \mathcal{L}\big(\mathbf{M}(q, \tilde{\mathbf O}_q), \mathbf{O}_q\big)
> > > $ (the optimization is over $\tilde{\mathbf O}_q$, Markdown here cannot display nested subscripts properly)
> > >     * Online LLM inference $y_u = \mathbf{M}(q, \mathbf{\tilde O}_q)$
> > >
> > > As formulated above, we emphasize that existing RAG-based unlearning methods operate only within the RAG module, thereby indirectly influencing the final LLM output. Their offline reverse-generation step does not incorporate the real-time query $q$ issued during inference. In contrast, UnRe treats the RAG and LLM components as a unified end-to-end system, directly optimizing the retrieved context $\mathbf{O}_q$ using the LLM's logits. This enables the model's output, conditioned on the current query $q$, to be explicitly steered toward the desired unlearning behavior.
> > >
> > > Since UnRe differs from prior RAG-based unlearning methods by performing query-specific context optimization, we draw inspiration from ECO [3] and adapt techniques from context-based attacks (e.g., H-ICL [11]) to perform inference-time gradient-based update for the unlearning setting. As shown in Eq. 5, rather than applying PGD directly, we introduce a constrained update rule on the forget piece ${o}_i$ to explicitly prevent the reintroduction or amplification of content that must be forgotten, while still allowing effective perturbation of the retrieved context. In summary, by leveraging these technical novelties, UnRe utilizes dynamic context to enable zero-shot continual unlearning—a critical capability that has been overlooked in prior work.

---

> > > ### Author Response · Authors · 2025-11-28
> > > **Response to Follow-up Questions (2/2)**
> > >
> > > **Q2: Follow-up on Q8: Alignment Details I am still unclear on the specific mechanism used to align the embedding model output with the frozen LLM's last-layer output. While it is true that encoder models (e.g., SimCSE [6], Sentence-BERT [7]) use the last transformer layer for representation, the latent spaces of an external embedding model and the target frozen LLM are likely different. How do you bridge this gap? Is there a specific projection or alignment technique used here?**
> > >
> > > Thank you for your follow-up question! We did not expand on this earlier because we do not consider the alignment component to be part of our core technical novelty. We used the alignment method from a prior work. The alignment process is outlined in Algorithm 2, and the xRAG alignment method [12] is cited in Section 3.2 of the main paper. The core idea is that the alignment module provides a learned projection that maps embeddings from the RAG encoder space into the LLM's embedding space.  The projector is trained with the retriever and LLM frozen by feeding the LLM with the *query embeddings* with the *projected retrieval embedding*, and jointly minimizing (i) a language-modeling loss to reconstruct the original context/answer and (ii) a distillation loss that makes its outputs match those of a full-text RAG teacher. After this projection, the transformed RAG embeddings can be treated as residing in the same latent space as the LLM’s input embeddings.
> > >
> > > We provide a detailed explanation of how the projection is performed in Appendix E.4.
> > >
> > >
> > >
> > > **Q3: Clarity and Re-review: I appreciate the authors' willingness to improve the writing and clarity based on feedback. However, a comprehensive re-review of the entire manuscript and notation is outside the scope of my current bandwidth. I also strongly encourage the authors to focus on sharpening the definitions of novelty and clearly articulating the new contributions, as I still find the distinctiveness of this work to be lacking in its current form.**
> > >
> > > Thank you for your thoughtful feedback. We will continue to carefully review the entire manuscript, with particular attention to clarity, notation consistency, and the precise articulation of our novelty and contributions. We will ensure that these revisions are fully reflected in the later version of the manuscript.
> > >
> > > Thank you again for your guidance.
> > >
> > >
> > >
> > > **References**
> > >
> > > [10] When Machine Unlearning Meets Retrieval-Augmented Generation (RAG): Keep Secret or Forget Knowledge? IEEE Transactions on Dependable and Secure Computing, 2025.
> > >
> > > [11] Hijacking Large Language Models via Adversarial In-context Learning. arXiv, 2023.
> > >
> > > [12] xRAG: Extreme Context Compression for Retrieval-augmented Generation with One Token. NeurIPS, 2024.

---

> ### Author Response · Authors · 2025-11-23
> **Rebuttal (2/4)**
>
> **W4: Efficiency: The method appears to require LLM generation, then RAG, then gradient updates, followed by another generation pass. This will likely slow inference substantially, which may make the approach impractical for real-world applications.**
>
> Thank you for pointing this out. We report the Time Complexity results for the experiments of Table 3 below, which are also included in Appendix G.3. Compared with ICUL [1], UnRe increases the per-query runtime by only about 4–5% (1.31 vs. 1.25). However, because UnRe requires no offline stage, its overall time for one epoch is significantly lower. Compared to LLMU [2], which relies heavily on an offline stage, UnRe achieves roughly a 44% reduction in total runtime.
>
> ### Time Complexity (seconds, averaged)
>
> | Method | Offline Total | Online Total | Overall for One Epoch|
> |--------|---------------|------------------|----------------------------|
> | LLMU   | 1684        | 493         | 2177  |
> | ICUL   | 317        | 534            | 851                     |
> | UnRe | 0             | 637           | 637                    |
>
> As discussed in the main paper, UnRe is designed for scenarios where the forget set changes frequently. To this end, UnRe is a 100% inference-time unlearning framework, which is extremely suitable for these scenarios by eliminating the long offline preparation stage at the cost of a slight overhead in inference.
>
>
> **Q1: Figure 1 is very difficult to interpret. The color coding is unclear, font sizes and families are inconsistent (even in the legend), and the flow alternates between left-to-right and right-to-left/top-to-bottom. The meaning of dashed versus solid arrows is not explained. It is also unclear where the LLM is involved: there is an arrow from the language model box to (3→4), but step 4 is labeled “retrieval” rather than “generation.” The figure does not clearly indicate which components are off-the-shelf (e.g., embedding model, LLM), and the legend uses vague categories such as “language model,” “embedding space,” and “offline preparation.” Overall, the figure looks unpolished and does not meet the clarity expected for an ICLR paper. Please revise for readability, consistent notation, and a coherent flow.**
>
> Thank you for your valuable comment. We have substantially revised Figure 1 based on your suggestions to improve clarity and consistency:
>
> 1) Unified the color scheme, font size, and font family.
> 2) Reorganized the overall flow into a clearer left-to-right, top-to-bottom structure.
> 3) Used solid arrows to represent communication between different components and dashed arrows to indicate state changes within the same component.
> 4) Clarified off-the-shelf modules (e.g., the embedding model and LLM) directly in the caption.
> 5) Simplified the legend to highlight only where each operation takes place (e.g., Contexts, Embedding Space).
> 6) Added an explicit explanation that Steps 2–5 form an iterative gradient-based update loop in the embedding space.
>
> Additionally, for better understanding, Section 3.2 has been updated to ensure a clearer alignment between the method overview and Figure 1.
>
> **Q2: Citations are not consistently formatted in LaTeX. In many places, citep should be used instead of citet. For example: “operates during LLM inference with frozen weights and is generally regarded as suppression-intended unlearning (Ren et al., 2025).”**
>
> Thanks for pointing this out. We used \cite for all the citations in the original submission. As you suggested, we have carefully revised the manuscript to ensure consistent use of \citep and \citet where appropriate. We appreciate your careful reading and helpful recommendation, which helps improve the clarity and correctness of the paper.
>
>
> **Q3: Line 167: What are the input x and label y in the forget piece? This is the first mention of labels in the paper. Why is a label needed here? Is x the content to forget? If so, why do we need y?**
>
> Thanks for your question! In our formulation, $x$ represents the input/query context, and $y$ represents the target content to forget (i.e., the ground-truth answer associated with $x$). So $y$ is the content that must be unlearned. As an example, in *Harry Potter* dataset, $x$ is *Harry Potter lives in*, and $y$ is *in 4 Privet Drive*. We follow the same notation as prior unlearning works such as ICUL [1] and ECO [3], which also use “input–label pairs ($x$,$y$)”.

---

> ### Author Response · Authors · 2025-11-23
> **Rebuttal (3/4)**
>
> **Q4: Line 171: The perturbed set \tilde{O}_q is introduced without sufficient context. What is its purpose at this stage, and how does it feature in the problem formulation**
>
> Thank you for the question. In Sections 3.1 and 3.2, we present the problem setup and the high-level workflow of UnRe. As stated in Section 3.1, the goal of UnRe is to find a perturbed context set $\tilde{O}_q$, which is then used as the retrieved context to guide the LLM toward an unlearned output. To avoid confusion with 'functions', we use $O$ instead of $F$ to denote the *overall* forget set, and $O_q$ to denote the *query-related* portion of that forget set retrieved at inference time. The perturbed version, $\tilde{O}_q$, is therefore the optimization target in our formulation, i.e., the context that UnRe iteratively updates so that the final LLM output meets the unlearning objective. We revised the text to clarify this role more explicitly.
>
>
> **Q5: Line 173: Using M.G to denote the LLM’s generation process is unconventional. Please align the notation with standard practice.**
>
> Thank you for the helpful suggestion. We originally wanted to use $\mathbf{M}.\mathcal{G}$ to avoid any misunderstanding that UnRe updates the weights or parameters of the LLM $\mathbf{M}$. Since UnRe operates entirely at inference time, we wanted to emphasize that we refer only to the *generation* process of the model with different inputs, not model training. Following your recommendation, we have simplified the notation to $\mathbf{M}()$ to better align with common practice.
>
>
> **Q6: Lines 186–188: The connection between performing gradient descent in V_R (the embedding model space) and imposing constraints on the LLM’s output distribution is not clear. Please provide a formal justification or an empirical rationale that links embedding-space updates to output distribution shifts.**
>
> Thank you for raising this important point. The link between embedding-space updates and output-distribution shifts has been investigated in prior attacking methods, such as soft prompt Threats [4], which demonstrate that small and targeted perturbations in the embedding layer can steer the model's output distribution. Several works on unlearning that we cite, including SPUL [5] and ECO [3], have also empirically demonstrated the connections between embedding updates and LLM output distribution shifts. UnRe adopts the same principle. We also included the details of embedding alignment in Alg. 2 of our original submission.
>
> We conduct additional experiments with different values of $\epsilon$, which can also be found in *Appendix H*:
>
> ### Performance with Different Values of $\epsilon$
>
> | Metric     | 0.00   | 0.05   | 0.10   | 0.15   | 0.20   | 0.30   | 0.40   | 0.50   | 0.60   |
> |------------|--------|--------|--------|--------|--------|--------|--------|--------|--------|
> | FQ Gap ↓   | 0.1503 | 0.1405 | 0.1331 | 0.1328 | 0.1331 | 0.1257 | 0.1225 | 0.1180 | 0.1177 |
> | PPL ↓      | 8.9520 | 8.9523 | 8.9527 | 8.9800 | 9.0277 | 9.1517 | 9.2866 | 9.4544 | 9.6291 |
>
> As shown by the results, increasing the perturbation magnitude of embedding-space updates will lead to stronger forgetting and gradual increases in perplexity. This provides direct evidence regarding the impact of the embedding-space perturbations on generation output.

---

> ### Author Response · Authors · 2025-11-23
> **Rebuttal (4/4)**
>
> **Q7: Line 196: φ is used to denote the embedding model’s forward pass, but the embedding model itself is not clearly specified or defined. Please clarify the model choice, training status, and interface.**
>
> Thank you for the helpful suggestion. We have updated Section 3.2 in the main paper to clearly describe the embedding model used in UnRe.
>
> In our experiments, UnRe relies on the same embedding model originally used in the model; we have updated it accordingly in Section 3.2. The model is pretrained and kept frozen throughout our method; we only use its forward pass to obtain token-level embeddings for context optimization.
>
>
> **Q8: Section 3.5: Why is the LLM hidden state h_δ(t) directly comparable to the embedding space produced by φ? The last hidden state of an LLM is typically optimized for next-token prediction rather than capturing the global semantics of the answer. Please justify this assumption or provide empirical evidence.**
>
> Thank you for the question. We follow similar practices as in prior works. Prior sentence-embedding and representation-learning works (e.g., SimCSE [6], Sentence-BERT [7]) use the last-layer hidden state as a semantic vector. In addition, recent activation-steering approaches (e.g., contrastive activation addition [8], InferAligner [9]) explicitly model the LLM hidden space as a linear semantic space in which directions correspond to interpretable behaviors. These results provide both conceptual and empirical evidence that $\bar h_\delta$ captures global semantics sufficiently.
>
> Besides, prior works such as Soft Prompt Threats [4] show that continuous embedding-space perturbations propagate through the frozen LLM and reliably modify later hidden states, even without updating any model parameters. The work shows that such embedding perturbations could propagate and induce shifts in hidden representations. This provides support that embedding-space updates, like those used in UnRe, have the capability of steering the hidden state of a frozen LLM toward an unlearned behavior.
>
>
> **Q9: Technical details on how the perturbation matrix is used in inference are missing. Is this matrix applied to all layers and all tokens of the transformer, or only to specific layers/states? Please provide a precise description of the application mechanism, scope, and computational overhead.**
>
> The perturbation matrix is applied only at the embedding layer and only to the retrieved context tokens. We have emphasized this in Section 3.5.1 of the revised manuscript. It is not applied to all layers. As described in Section 3.2, UnRe generates the perturbed context set $\tilde{O}_q$, and the optimization is carried out directly in the embedding space. We have updated the description in Section 3.2 to clearly state that the perturbation matrix $\delta$ is the central component driving the optimization and that the entire perturbation procedure takes place at the embedding layer.
>
> Regarding computational overhead, we have already reported the overall time complexity in our earlier response (W4). Below, we provide an additional breakdown showing the time consumption of each component to further clarify where the computational cost arises.
>
> | Step | Component  | Time Breakdown |
> |----|-----------------------------------------|--------------------------------|
> | 1  | Retrieval & Embedding with Membership Inference       | 4.6%  |
> | 2  | Embedding-level Gradient Optimization        | 55.7%  |
> | 3  | Re-decoding with Perturbed Context      | 39.7%  |
>
>
>
>
> **References**
>
> [1] In-context Unlearning: Language Models as Few-shot Unlearners. ICML, 2024.
>
> [2] Large Language Model Unlearning. NeurIPS, 2024.
>
> [3] Large Language Model Unlearning via Embedding-Corrupted Prompts. NeurIPS, 2024.
>
> [4] Soft Prompt Threats: Attacking Safety Alignment and Unlearning in Open-source LLMs through the Embedding Space. NeurIPS, 2024.
>
> [5] Soft Prompting for Unlearning in Large Language Models. NAACL, 2025.
>
> [6] Simple Contrastive Learning of Sentence Embeddings. EMNLP, 2021.
>
> [7] Sentence Embeddings using Siamese BERT-Networks. EMNLP-IJCNLP, 2019.
>
> [8] Steering Llama2 via Contrastive Activation Addition. ACL, 2024.
>
> [9] Inference-Time Alignment for Harmlessness through Cross-Model Guidance. EMNLP, 2024.

---

### Official Review · Reviewer_wMLX · 2025-10-31

**Soundness:** 3
**Presentation:** 3
**Contribution:** 2
**Rating:** 6
**Confidence:** 4

**Summary:**

The paper introduces UNRE, a framework for zero-shot, inference-time unlearning in large language models (LLMs) using dynamic, query-conditioned contextual retrieval. Unlike previous methods requiring retain sets or offline-trained modules, UNRE operates solely on a forget set and employs retrieval-augmented generation (RAG) with online membership inference to identify, adapt, and perturb context embeddings in response to real-time queries, aiming to suppress memorized or undesirable content while retaining overall model utility

**Strengths:**

1. The proposed method transforms the unlearning problem into a context-level intervention, avoiding the costly retraining required for parameter-level unlearning.

2. The theoretical and algorithmic design is simple yet effective, achieving context switching between forgetting and retention through embedding-space perturbation.

3. UNRE is evaluated on multiple public benchmarks, including entity unlearning and copyright unlearning, demonstrating strong experimental performance.

**Weaknesses:**

1. The method uses a similarity threshold ( \tau ) to decide whether to trigger unlearning based on the similarity between the model output ( y_q ) and the forget set ( O ). Since ( \tau ) is the key factor determining whether unlearning occurs, treating this decision as a binary classification may be oversimplified. Its value can be sensitive to the model, task, and data, and the paper does not explain how it is chosen.

2. The loss ( \mathrm{softplus}(N - S) ) aims to balance semantic preservation (high ( S )) and output divergence (low ( N )), which are conflicting goals. If ( S ) dominates, forgetting may be incomplete; if ( N ) dominates, the output may drift semantically. A tunable hyperparameter could help balance these effects.

3. The Projected Gradient Descent (PGD) optimization may fall into local minima or become unstable in high-dimensional embedding spaces. Exploring contrastive-learning-based optimization or low-dimensional perturbation approximations could improve stability.

4. Each unlearning step requires document retrieval, multiple PGD iterations, and re-decoding, which significantly increases inference latency. This may limit the method’s practicality in real-time applications.

5. As a context-level approach, the method may be more vulnerable to jailbreak or adversarial attacks, where an attacker could craft prompts to bypass unlearning and recover forgotten content.

**Questions:**

See weaknesses.

---

> ### Author Response · Authors · 2025-11-23
> **Rebuttal (1/2)**
>
> Thank you for your valuable suggestions and insightful comments. We have addressed all the concerns and revised the paper (highlighted in blue) accordingly. Please kindly see our response to each comment below.
>
>
> **W1: The method uses a similarity threshold ( \tau ) to decide whether to trigger unlearning based on the similarity between the model output ( y_q ) and the forget set ( O ). Since ( \tau ) is the key factor determining whether unlearning occurs, treating this decision as a binary classification may be oversimplified. Its value can be sensitive to the model, task, and data, and the paper does not explain how it is chosen.**
>
> Thank you for pointing this out. Our use of $\tau$ is consistent with prior works, for example, ECO [1]. We also set the same $\tau$ threshold in our experiments for fair comparison. In practice, $\tau$ can be set differently based on the application scenarios, as there is an inevitable trade-off between forgetting and generation performance with different values of $\tau$.
>
> Following the suggestion, we conduct additional experiments on the sensitivity analysis of $\tau$, which is also added into *Appendix H* of the revised manuscript.
>
> ### Sensitivity to $\tau$ ($\epsilon$ = 0.10)
>
> | Metric     | 0.30   | 0.50   | 0.70   | 0.85   | 0.90   | 0.92   | 0.94   | 0.95   | 0.96   | 0.97   | 0.98   | 0.99   |
> |------------|--------|--------|--------|--------|--------|--------|--------|--------|--------|--------|--------|--------|
> | FQ Gap ↓   | 0.0520 | 0.0735 | 0.0914 | 0.1082 | 0.1207 | 0.1245 | 0.1280 | 0.1331 | 0.1518 | 0.1752 | 0.2011 | 0.2317 |
> | PPL ↓      | 10.6414  | 10.1235  | 9.7306  | 9.2101  | 8.9526  | 8.9534  | 8.9544  | 8.9539  | 8.9550  | 8.9603  | 8.9657  | 8.9722  |
>
> **W2: The loss $( \mathrm{softplus}(N - S) )$ aims to balance semantic preservation (high ( S )) and output divergence (low ( N )), which are conflicting goals. If ( S ) dominates, forgetting may be incomplete; if ( N ) dominates, the output may drift semantically. A tunable hyperparameter could help balance these effects.**
>
> Many thanks for your suggestion! As it was mentioned in the main paper, our loss design is inspired by prior works including ECO [1], FLAT [2], and LLMU [3], and the goal is to *maintain the semantic meanings (high $S$) while increasing token distributional shift (low $\mathcal{N}$)*. Meanwhile, our empirical results show that $\operatorname{softplus}\!\big(\mathcal{N} - S\big)$ already provides a favorable forget–retain balance in our experimental results. So we did not add an additional coefficient. But we agree that adding a tunable hyperparameter can help balance these two losses in different scenarios. We updated that to a more general form $\operatorname{softplus}\!\big(\lambda \mathcal{N} - S\big)$ in the revised manuscript. We also perform an additional ablation study on $\lambda$, as shown in Figure 3 of *Appendix I*. It can be seen that $\lambda$ is best set in the range of $[0,3]$ and $\lambda=1$ yields decent performance. Therefore, we retain the remaining experimental results with $\lambda=1$.

---

> ### Author Response · Authors · 2025-11-23
> **Rebuttal (2/2)**
>
> **W3: The Projected Gradient Descent (PGD) optimization may fall into local minima or become unstable in high-dimensional embedding spaces. Exploring contrastive-learning-based optimization or low-dimensional perturbation approximations could improve stability.**
>
> We sincerely appreciate your insightful suggestions. Our experimental results generally show stable behavior, but we agree that first-order method can become unstable in high-dimensional embedding spaces. There is clear room for further improvement. We view contrastive-learning–based optimization and low-dimensional perturbation approximations as promising future directions, with strong potential to further enhance the performance and robustness of the proposed UnRe framework.
>
>
> **W4: Each unlearning step requires document retrieval, multiple PGD iterations, and re-decoding, which significantly increases inference latency. This may limit the method’s practicality in real-time applications.**
>
> Thank you for pointing this out. We report the Time Complexity results for the experiments of Table 3 below, which are also included in Appendix G.3. Compared with ICUL [5], UnRe increases the per-query runtime by only about 4–5% (1.31 vs. 1.25). However, because UnRe requires no offline stage, its overall time for one epoch is significantly lower. Compared to LLMU [4], which relies heavily on an offline stage, UnRe achieves roughly a 44% reduction in total runtime.
>
> ### Time Complexity (seconds, averaged)
>
> | Method | Offline Total | Online Total | Overall for One Epoch|
> |--------|---------------|------------------|----------------------------|
> | LLMU   | 1684        | 493         | 2177                     |
> | ICUL   | 317        | 534            | 851                     |
> | UnRe | 0             | 637           | 637                    |
>
> As discussed in the main paper, UnRe is designed for scenarios where the forget set changes frequently. To this end, UnRe is a 100% inference-time unlearning framework, which is extremely suitable for these scenarios by eliminating the long offline preparation stage at the cost of a slight overhead in inference.
>
>
> **W5: As a context-level approach, the method may be more vulnerable to jailbreak or adversarial attacks, where an attacker could craft prompts to bypass unlearning and recover forgotten content.**
>
> Thanks for your insights! Yes, attacking unlearning is an interesting direction. We would like to kindly note that jailbreak or adversarial attacks are not typically evaluated in prior works, including ECO [1], ICUL [3], GUARD [4], and SPUL [6].
>
> Because UnRe operates at the context level, it reduces exposure to query-level manipulations that attempt to elicit forgotten information. We conducted another review of the literature during the rebuttal period, but we did not find a well-established or widely adopted attack framework specifically aimed at inference-time unlearning. For context-level attacks, techniques such as H-ICL [7] have been studied, but they primarily serve to perturb contextual evidence rather than recover content intentionally suppressed by unlearning methods. Interestingly, because UnRe adapts optimization ideas from strong context-perturbation methods, reversing such strategies to attempt recovering forgotten content might be a potentially effective attack.
>
>
> **References**
>
> [1] Large Language Model Unlearning via Embedding-Corrupted Prompts. NeurIPS, 2024.
>
> [2] LLM Unlearning via Loss Adjustment with Only Forget Data. ICLR, 2025.
>
> [3] Large Language Model Unlearning. NeurIPS, 2024.
>
> [4] GUARD: Generation-time LLM Unlearning via Adaptive Restriction and Detection. arXiv, 2025.
>
> [5] In-context Unlearning: Language Models as Few-shot Unlearners. ICML, 2024.
>
> [6] Soft Prompting for Unlearning in Large Language Models. NAACL, 2025.
>
> [7] Hijacking Large Language Models via Adversarial In-context Learning. arXiv, 2023.

---

> > ### Comment · Reviewer_wMLX · 2025-11-24
> >
> > Thanks for your response to answering my questions. I will keep my score.

---

> > > ### Author Response · Authors · 2025-11-28
> > >
> > > Thank you very much for your valuable comments and support. If you have any additional questions, please don't hesitate to let us know. We would be happy to address them promptly.

---

### Official Review · Reviewer_qutX · 2025-11-01

**Soundness:** 2
**Presentation:** 1
**Contribution:** 2
**Rating:** 4
**Confidence:** 4

**Summary:**

This paper proposes UNRE, a zero-shot inference-time unlearning framework for large language models (LLMs). Unlike prior approaches that require retain sets, offline fine-tuning, or fixed unlearning contexts, UNRE leverages query-adaptive dynamic retrieval based on the forget set only. Specifically, it integrates an online membership inference module to identify query-related forget pieces, then applies gradient-based refinement of retrieved embeddings to steer model outputs toward an “unlearned” distribution.

**Strengths:**

- The paper explores a new setting of zero-shot inference-time unlearning with dynamic retrieval, extending prior in-context unlearning (ICUL) work.

- Unlike typical in-context learning methods, it operates at the embedding level to prevent the activation of information associated with the forget set.

- The motivation is well-aligned with privacy and copyright unlearning, and the approach explicitly avoids model parameter modification, retain data, or retraining, making it highly practical for real-world deployment scenarios.

**Weaknesses:**

- The paper could benefit from qualitative examples showing cases where UNRE fails (e.g., borderline membership detection, overly aggressive context updates) to better interpret its robustness.

- Some improvements (e.g., small PPL changes) might result from threshold or step-size tuning in Algorithm 1. Sensitivity analyses on τ and ε would clarify stability and generalization.

- The embedding-level perturbation may risk distorting semantic representations for complex queries. While PGD constraints are mentioned, an explicit analysis of semantic preservation vs. forgetting trade-off is missing.

**Questions:**

1. There is an incorrect bolded value in Table 1 under the Falcon3-7B-Instruct setting, where Prompt FQ (0.0970) is actually much higher than UnRE (0.0611). In addition, for the same setting, it is unclear why the MU score of UnRE (0.0644) is significantly lower than that of the other methods  (0.66).

2. Could you please bold the best value in Table 2 for clarity?

3. How sensitive is UNRE to the similarity threshold (τ) and update budget (ε)?

4. Could the authors provide ablation studies showing the contribution of each component — membership inference, gradient-based update, and semantic loss — to the overall unlearning performance?

---

> ### Author Response · Authors · 2025-11-23
> **Rebuttal (1/2)**
>
> Thank you very much for your thoughtful suggestions and insightful feedback. We have made revisions (highlighted in blue) to address all the points raised. Our detailed responses to each comment are provided below.
>
>
> **W1: The paper could benefit from qualitative examples showing cases where UnRe fails (e.g., borderline membership detection, overly aggressive context updates) to better interpret its robustness.**
>
> We greatly appreciate for your thoughtful suggestions! We exhibit 3 failed scenarios with selected examples. Please refer to the blue text in *Appendix J*. We also present the failure rate for each category below, showing that UnRe has a low failure rate.
>
> | Failure type                          | Failure rate |
> |---------------------------------------|------------------------------|
> | F1: Borderline membership detection   | 1.1%                      |
> | F2: Overly aggressive context update  | 3.4%                      |
> | F3: Insufficient forgetting / leakage | 1.8%                      |
> | **Total**                             | **6.3%**    |
>
>
> **W2: Some improvements (e.g., small PPL changes) might result from threshold or step-size tuning in Algorithm 1. Sensitivity analyses on τ and ε would clarify stability and generalization.**
>
> Following your valuable suggestions, we show the sensitivity analysis results of $\tau$ and $\epsilon$, which are also added in *Appendix H* of the revised manuscript.
>
> ### Sensitivity to $\tau$ ($\epsilon$ = 0.10)
>
> | Metric     | 0.30   | 0.50   | 0.70   | 0.85   | 0.90   | 0.92   | 0.94   | 0.95   | 0.96   | 0.97   | 0.98   | 0.99   |
> |------------|--------|--------|--------|--------|--------|--------|--------|--------|--------|--------|--------|--------|
> | FQ Gap ↓   | 0.0520 | 0.0735 | 0.0914 | 0.1082 | 0.1207 | 0.1245 | 0.1280 | 0.1331 | 0.1518 | 0.1752 | 0.2011 | 0.2317 |
> | PPL ↓      | 10.6414  | 10.1235  | 9.7306  | 9.2101  | 8.9526  | 8.9534  | 8.9544  | 8.9539  | 8.9550  | 8.9603  | 8.9657  | 8.9722  |
>
> ### Sensitivity to $\epsilon$ ($\tau$ = 0.95)
>
> | Metric     | 0.00   | 0.05   | 0.10   | 0.15   | 0.20   | 0.30   | 0.40   | 0.50   | 0.60   |
> |------------|--------|--------|--------|--------|--------|--------|--------|--------|--------|
> | FQ Gap ↓   | 0.1503 | 0.1405 | 0.1331 | 0.1328 | 0.1331 | 0.1257 | 0.1225 | 0.1180 | 0.1177 |
> | PPL ↓      | 8.9520 | 8.9523 | 8.9527 | 8.9800 | 9.0277 | 9.1517 | 9.2866 | 9.4544 | 9.6291 |
>
> Overall, the results show that UnRe is not overly sensitive to these hyperparameters within a reasonable range:
>
> * For $\tau$, increasing the threshold gradually strengthens forgetting (lower FQ Gap), while the PPL remains stable across a broad interval. Note that prior works such as ECO [1] also use $\tau$ in the same range.
> * For $\epsilon$, larger perturbation budgets allow slightly stronger forgetting, with only mild degradation in PPL. Once $\epsilon$ is within a moderate range (i.e., >0.1), further increases yield diminishing returns.
>
> These results confirm that the observed performance improvements are not artifacts of fragile hyperparameter choices, and that UnRe remains stable under a reasonable range of $\tau$ and $\epsilon$ values.
>
>
> **W3: The embedding-level perturbation may risk distorting semantic representations for complex queries. While PGD constraints are mentioned, an explicit analysis of the semantic preservation vs. forgetting trade-off is missing.**
>
> Many thanks for your suggestion. We'd like to clarify that our perturbation is added only to the context retrieved by UnRe and is used to guide the LLM in generating unlearned output. Thus, the semantic preservation towards the context is not a primary goal. Meanwhile, UnRe does not add any perturbations towards the original query.
>
> For evaluating the semantic preservation of the LLM generation, we have presented the results through *Forgetting(unlearning) vs. Semantic Preservation* in different benchmarks, for example, in Table 1, *FQ and F-RL* vs. *MU and R-RL*; in Table 2, *Forget* vs. *Neighbor and Utility*; and in Table 3, *FQ Gap* vs. *PPL and Avg. ACC*. These results demonstrate the trade-off between unlearning performance and semantic preservation.
>
> We have also conducted an additional ablation study as suggested. *Appendix H* show how $\epsilon$ is influencing the performance of semantic preservations.

---

> ### Author Response · Authors · 2025-11-23
> **Rebuttal (2/2)**
>
> **Q1: There is an incorrect bolded value in Table 1 under the Falcon3-7B-Instruct setting, where Prompt FQ (0.0970) is actually much higher than UnRE (0.0611). In addition, for the same setting, it is unclear why the MU score of UnRE (0.0644) is significantly lower than that of the other methods (0.66).**
>
> Thank you for pointing this out, and we apologize for the mistake. We have re-run the experiment and confirmed that the MU score was indeed a typo. The corrected value (0.6644) has now been updated. We have also corrected the boldfacing: the Prompt result for MU on Falcon-3 is now properly highlighted, and all remaining bolded values correctly represent the best scores.
>
> We have also carefully checked all the results in the paper.
>
>
> **Q2: Could you please bold the best value in Table 2 for clarity?**
>
> We appreciate your suggestion. Please kindly see the updated Table 2, and we have emphasized the best values in the tables of the main paper. Please note that the primary goal of UnRe is to develop a 100% inference-time unlearning framework that can achieve strong performance for unlearning while maintaining decent semantic preservation.
>
>
> **Q3: How sensitive is UNRE to the similarity threshold ($\tau$) and update budget ($\epsilon$)?**
>
> Please kindly refer to our response to W2 above.
>
>
> **Q4: Could the authors provide ablation studies showing the contribution of each component — membership inference, gradient-based update, and semantic loss — to the overall unlearning performance?**
>
> Thanks for your suggestion. We have conducted additional experiments to evaluate the contribution of each component.
>
> | Components | FQ Gap ↓ | PPL ↓   | Avg. Acc. ↑ |
> |-----------|----------|---------|-------------|
> | Full UnRe               | 0.1207 | 8.9524 | 0.5617 |
> | w/o membership inference gate      | 0.1150 | 9.0501 | 0.5521 |
> | w/o gradient-based update (no PGD) | 3.6507 | 8.9524 | 0.5617 |
> | w/o semantic loss term             | 0.0911 | 9.2506 | 0.5483 |
>
> For *w/o membership inference gate*, we set $\tau=1$; for *w/o gradient-based update (no PGD)*, we adopt Hotflip [2] as a replacement; for *w/o semantic loss term*, we remove the semantic loss in the loss function. It can be seen that there would be a considerable performance degradation if we remove any of the components. These ablations confirm that all three components—membership inference, gradient-based updates, and the semantic loss—contribute complementary benefits to the overall unlearning performance. The results align with our expectations. Each component plays a distinct and essential role in enabling UnRe to perform effective inference-time unlearning:
>
> * Membership inference identifies the query-relevant knowledge the LLM was originally trained on and must forget. Without this step, UnRe cannot determine which portions of the model’s parametric memory should be suppressed, resulting in incomplete or failed unlearning.
> * Gradient-based embedding updates provide a controllable mechanism for steering the output distribution away from the forgetting target. Without the gradient step, UnRe degenerates into simple context passing, losing its ability to meaningfully influence or reshape the model’s generation.
> * Semantic loss constrains the optimization to preserve coherence and answer quality. Removing this term increases forgetting but leads to degraded fluency and correctness, demonstrating that semantic regularization is necessary to maintain utility while optimizing for unlearning.
>
> We have clarified this in the revision by adding the corresponding experimental results and discussion to *Appendix I.1*.
>
>
> **References**
>
> [1] Large Language Model Unlearning via Embedding-Corrupted Prompts. NeurIPS, 2024.
>
> [2] HotFlip: White-box Adversarial Examples for Text Classification. ACL, 2018.

---

### Author Response · Authors · 2025-11-23
**Thank you for your reviews!**

Dear Reviewers,

Thank you again for the time and effort you put into reading our manuscript and for providing thoughtful, constructive feedback. Your comments greatly helped us improve the clarity and quality of the paper. We hope that our responses have addressed your concerns.

We appreciate your consideration and are happy to engage in any further discussion.

Sincerely,
Authors of UnRe

---

### Author Response · Authors · 2025-12-04
**Rebuttal Summary**

Dear Area Chairs and Senior Area Chairs,

We sincerely appreciate the time and effort of the area chairs in evaluating our work. We also thank all reviewers for their thoughtful, constructive, and highly valuable feedback. We have carefully addressed every concern raised across the reviews. Below, we provide a concise summary of our responses and the key revisions incorporated in the updated manuscript (all changes are marked in blue).

**1. Clarification of Core Novelty Compared to Prior RAG-Based Unlearning Works**

We clarified our novelty extensively during the rebuttal:

Prior RAG-unlearning methods rely on offline-generated retain sets or rewriting modules whose content is fixed during inference and cannot adapt to each query. UnRe is the first fully inference-time unlearning framework designed for scenarios where forget sets change frequently. Unlike prior methods that act only inside the RAG module, UnRe treats RAG-LLM as a single system, directly optimizing the retrieved context using LLM logits to steer generation toward unlearning.

We also added a side-by-side workflow comparison, a concrete illustrative example, and a more formal formulation to emphasize this distinction.

**2. Notation, Clarity, and Exposition Improvements**

All unclear variables, notation, and definitions raised by reviewers were revised through reorganization of Sections 3.1–3.2 and clarification of the problem setup.

We corrected all citation formatting, rewrote ambiguous explanations, and also substantially revised Figure 1 for clarity.

**3. Embedding–LLM Alignment**

We clarified that the alignment module follows prior work (xRAG) and is not part of UnRe's novelty. We provided a detailed explanation (clarifying the original Algorithm 2 and adding Appendix E.4) of how the embedding alignment is applied.

**4. Empirical Justification of Gradient-Based Context Optimization**

To address concerns about the impact of embedding-space perturbations, we added new sensitivity analyses for both $\tau$ and $\epsilon$.

Meanwhile, we conducted new ablation studies to isolate the contributions of different components: membership inference, gradient updates, and semantic loss.

We provided details on how embedding-space updates shift the output distribution (Appendix H).

**5. Efficiency**

We added time complexity comparisons between UnRe and prior works such as LLMU and ICUL, as well as a breakdown of computation costs by component. We clarified that UnRe only incurs a ~4–5% increase in online runtime compared to ICUL, while eliminating the high offline cost.

**6. Failure Case Analysis**

In the revised version, we added three categories of failures (borderline detection, over-modification, and insufficient forgetting) along with their corresponding failure rates.

**7. Semantic Preservation vs. Forgetting**

We clarified how semantic preservation is evaluated in the original paper, and added further justification through new ablations.



**Final Note**

We appreciate the reviewers’ detailed and insightful comments. We have thoroughly revised the paper and addressed all the concerns during the rebuttal period.  The new results and discussions further highlight UnRe's novelty, effectiveness, practical relevance, and broad applicability.

Overall, UnRe introduces a new perspective on LLM unlearning by showing, for the first time, that dynamic, query-adaptive context optimization at inference time can steer a frozen model toward unlearned behaviors. Methodologically, our framework unifies retrieval and generation into an end-to-end system: a lightweight similarity-based membership inference gate identifies queries tied to forget content, and a constrained gradient update is then applied only to the retrieved forget-context embeddings to suppress forgotten knowledge without modifying the query or model parameters. This design eliminates all offline costs, which is distinctive from prior RAG-based approaches that also cannot adapt to new forget sets or new queries in real-time. Taken together, UnRe offers a practical and lightweight solution for real-world dynamic unlearning scenarios, advancing the methodological foundations of inference-time LLM unlearning.


Sincerely,

Authors of UnRe

---

### Meta-Review · Area_Chair_VwJZ · 2026-01-07

**Summary:**

This paper proposes UnRe, an inference-time LLM unlearning framework. The method first identifies forget set based on user input query, then dynamically updates the retrieved embeddings in a zero-shot setting, steering the model toward a query-specific unlearned distribution.

Before rebuttal, reviewers hold divergent opinions toward the paper. Reviewers acknowledge the importance of efficient inference-time unlearning and the benefits of avoiding parameter updates (qutX, wMLX, M4qg). The idea of dynamically perturbing retrieved context embeddings is also viewed as interesting (wMLX, M4qg). However, reviewers raise several concerns. The major concerns are as follows:
* Novelty. Reviewer M4qg questioned whether the proposed method meaningfully differs from prior RAG-based unlearning approaches, and was not fully convinced that inference-time context optimization constitutes a substantial algorithmic advance.
* Clarity and technical detail. Reviewer M4qg raises strong concerns about unclear notation, missing technical details, and an unpolished writing, which hinder understanding of the method.
* Efficiency and robustness. Reviewer wMLX and M4qg note increased inference-time overhead. Besides, reviewer wMLX raised concerns about potential sensitivity to hyperparameters, and the stability of PGD-based optimization.
* Experimental analysis. Reviewer qutX requests additional ablations, sensitivity analyses, and qualitative failure cases to better assess robustness.

The authors made substantial efforts during the rebuttal and addressed many of the major concerns. However, after rebuttal, there is still potential concerns regarding novelty, ablation results, and technical details. The resolved and remaining issues are discussed in the following section.

**Reviewer Concerns:**

During rebuttal, many major concerns are resolved, while some important concern remain. Specifically:
* Efficiency, robustness, experimental analysis: The authors provide additional experiments that resolve most of these concerns. In particular, the time complexity analysis shows modest inference-time overhead but substantially reduced offline computation, and the sensitivity studies indicate that UnRe is stable under a reasonable range of hyperparameter settings. The authors also include ablation studies and qualitative failure cases. However, the ablation results do not clearly demonstrate the necessity of the membership inference gate, as its removal yields performance comparable to the full UnRe. This weakens the empirical justification for the contribution of this module.
* Novelty: There has been a detailed discussion between the authors and reviewer M4qg regarding novelty. The key question is how UnRe differs from traditional RAG-based unlearning approaches. While reviewer M4qg raised additional concerns after the first rebuttal, the authors’ follow-up response clarifies that the main distinction is that UnRe performs context optimization at inference time, whereas prior RAG-based unlearning methods do not modify retrieved contexts during inference. However, it is important to note that traditional RAG-based approaches already retrieve contexts conditioned on the input query. Therefore, the “query-specific” feature, which is listed as a contribution bullet in the introduction, risks being overstated. As a result, the current framing blurs the core novelty of UnRe.
* Clarity and technical detail: Reviewer M4qg finds that the original submission lacks clarity and sufficient technical detail, and raises a series of detailed questions regarding notation and formulation. The authors make substantial efforts in the rebuttal to address these issues and revise the presentation. However, reviewer M4qg notes that these revisions constitute major changes to the original text. While this issue alone may be minor, combining with the novelty concerns, the paper would benefit from an additional round of polishing.

**Reviewer Scores:**

Reviewer wMLX explicit states to remain the original score (6 -> 6).

Reviewer qutX: Most of the concerns are resolved during rebuttal. One remaining issue is that the ablation does not clearly demonstrate the necessity of the membership inference gate. The score could reasonably remain 4 or increase to 6.

Reviewer M4qg: The reviewer remains conservative throughout the rebuttal phase and does not change the original score after discussing with the author, maintaining a score of 2. However, the AC checks the response regarding the novelty concern and acknowledges the algorithmic difference between UnRe and traditional RAG-based unlearning methods. On the other hand, the query-specific novelty risks being overstated. If reviewer M4qg fully participates in the discussion, the score could remain 2 or increase to 4.

---

### Decision · Program_Chairs · 2026-01-26

Reject